# Recurrence along Depth: Deep Convolutional Neural Networks with Recurrent Layer Aggregation

**Jingyu Zhao, Yanwen Fang and Guodong Li**
Department of Statistics and Actuarial Science
The University of Hong Kong
{gladys17, u3545683}@connect.hku.hk, gdli@hku.hk

## Abstract

This paper introduces a concept of layer aggregation to describe how information from previous layers can be reused to better extract features at the current layer. While DenseNet is a typical example of the layer aggregation mechanism, its redundancy has been commonly criticized in the literature. This motivates us to propose a very light-weighted module, called recurrent layer aggregation (RLA), by making use of the sequential structure of layers in a deep CNN. Our RLA module is compatible with many mainstream deep CNNs, including ResNets, Xception and MobileNetV2, and its effectiveness is verified by our extensive experiments on image classification, object detection and instance segmentation tasks. Specifically, improvements can be uniformly observed on CIFAR, ImageNet and MS COCO datasets, and the corresponding RLA-Nets can surprisingly boost the performances by 2-3% on the object detection task. This evidences the power of our RLA module in helping main CNNs better learn structural information in images.

## 1 Introduction

Convolutional neural networks (CNNs) have achieved notable success in computer vision tasks, crediting to their ability to extract high-level features from input images. Due to rapid growth in the depth of CNNs in recent years, the problem of how to pass information efficiently through layers often arises when designing deep architectures. Residual connections [20, 21], or skip connections, are now cornerstone components that act as information pathways to give layers direct access to previous ones and make training feasible with hundreds of layers. In a deep feature hierarchy, higher-level features learned by the network are built upon simpler ones, but not necessarily on the layer right before it [17]. This conjecture can be supported by the stochastic depth training method [27], where layers randomly gain access to previous ones. Furthermore, it is discovered that entire layers can be removed from ResNets without impacting the performance. This observation incubated the dense connectivity in DenseNets [28, 29], where layers in a stage have direct access to all previous layers through fully connected skip connections.

We introduce a concept of layer aggregation to systematically study network designs on feature reuse, and a typical example is DenseNet. However, within a densely-connected stage, the number of network parameters grows quadratically with respect to the number of layers. These highly redundant parameters may extract similar features multiple times [7], and limit the number of channels to store new information. This motivates us to propose a very light-weighted recurrent layer aggregation (RLA) module with much fewer parameters. Specifically, we use recurrent connections to replace dense connectivity and achieve a parameter count independent of the network depth. The RLA

35th Conference on Neural Information Processing Systems (NeurIPS 2021).

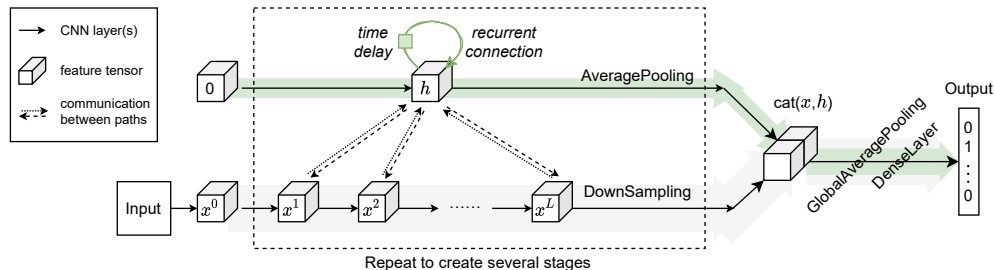

Figure 1: Schematic diagram of a CNN with recurrent layer aggregation for image classification.

module preserves the layer aggregation functionality, and it can be added to existing CNNs for better feature extraction.

Our RLA module is compatible with many CNNs used today. A schematic diagram is provided in Figure 1, where $x$ represents hidden layers in a deep CNN and $h$ represents hidden states in RLA modules. LeCun et al. [33] pointed out that unfolded RNNs can be seen as very deep feedforward networks where all the hidden layers share the same weights. When examined alone, an RLA module behaves similar to an unfolded RNN with layers of a deep CNN as its inputs. But applying RLA modules to CNNs is more than a simple combination of RNN and CNN (e.g., [41]). As the key idea behind layer aggregation is to provide the current layer with a version of layer history, information exchange between main CNN and RLA modules is a must. This results in a connection from the hidden unit of the RNN back to its inputs, which is hardly observable in other deep architectures. Moreover, we do not use global average pooling when passing information from the main CNN to RLA, so that the historical information is enriched and contains spatial information.

Despite its recurrent design, a convolutional-RNN-based RLA module can be easily implemented using standard convolutions with parameter sharing. Empirically, RLA modules are computationally light-weighted and impose only a slight increase in model parameters and computational cost. In terms of functionality, it serves purposes beyond channel attention and can be applied on top of channel attention modules. We perform extensive experiments across a variety of tasks using different deep CNN architectures, including ResNets [20, 21], ECANets [54], Xception [10] and MobileNetV2 [43]. Our experimental results show that RLA consistently improves model performances on CIFAR, ImageNet and MS COCO datasets. On MS COCO, our RLA module can achieve 2-3% gains in terms of average precision, which significantly outperforms other state-of-the-art networks with remarkable performances.

The main contributions of our work are below: (1) We introduce a definition of layer aggregation for analyzing CNN architectures. (2) We propose a novel recurrent layer aggregation module, with motivation from the layer aggregation mechanism and time series analysis. (3) We investigate detailed RLA module designs through ablation study and provide guidelines on applying RLA modules. (4) We show the effectiveness of RLA across a broad range of tasks on benchmark datasets using several popular deep CNN architectures.

## 2    Related work

**CNN-RNN hybrid models**    Deep CNNs with RLA modules should be easily distinguishable from other CNN-RNN hybrid networks, as RLA features information exchange between CNN and RNN at each layer and the corresponding recurrent update. In visual description tasks, RNNs are adopted to process the outputs of CNNs for sequence generation [14, 51, 44, 53, 5]. For video super-resolution, convolutional LSTMs can be used first to capture the temporal dependence between frames and then pass outputs to CNNs [38]. There are only a few papers that apply the concept of recurrence to CNNs. In RecResNet [1], a whole CNN is used recurrently for disparity map refinement. Moreover, in [36], convolutional layers are used recurrently to produce "deep" CNNs before the debut of ResNets. Dense-and-Implicit Attention Network [30] possesses a similar high-level CNN-RNN combination structure compared with ours, while their design is within the attention framework.

**CNNs with branching paths**    CNNs with our RLA module have similar structures compared to those with branching paths, e.g., Inception networks [46, 31, 47, 48], or networks with grouped convolutions, e.g., AlexNet [32] and ResNeXt [56]. The main difference is that our RLA module has

shared convolutional kernels along depth. When RLA is applied to ResNets, the resulting networks can be viewed as compressed Dual Path Networks [7].

**Attention mechanisms in CNNs** Attention mechanisms have been incorporated into CNNs mainly in the form of spatial attention and channel attention. Successful applications include image captioning [57, 5], visual question answering [62], image classification [52, 16], semantic segmentation [34], face verification [6], and video frame interpolation [9]. An outstanding design of channel attention would be the Squeeze-and-Excitation (SE) block [26], which is later adopted by MobileNetV3 [25] and EfficientNet [49] and developed into compressed variants [54]. The proposed layer aggregation mechanism has a connection with channel attention across multiple layers, and meanwhile, it can be applied to networks with channel attention modules.

**RNN variants** Since RLA modules adopt the form of a convolutional RNN, RNN variants give rise to different RLA designs. For example, the hidden feature maps in RLA can also be updated by the convolutional counterparts of LSTM [24, 45] or GRU cells[8]. In terms of architectural connectivity, skip connections can be introduced to RNN hidden states [40, 60, 3, 13]. Moreover, for deep RNNs, different layers of state variables can be updated at different frequencies [15]. Correspondingly, we may derive deep RLA modules that formulate stage aggregation on top of layer aggregation.

# 3 Recurrent layer aggregation modules

This section first introduces a concept of layer aggregation and some parsimonious models in time series analysis, which together motivate a type of light-weighted recurrent layer aggregation (RLA) modules by making use of the sequential structures of deep CNNs.

## 3.1 Layer aggregation

Consider a deep CNN with $x^t$ being the hidden features at the $t$th layer and $x^0$ being the input, where $L$ is the number of layers, and $1 \leq t \leq L$. The CNN is said to have the *layer aggregation mechanism* if

$$A^t = g^t(x^{t-1}, x^{t-2}, \ldots, x^0) \quad \text{and} \quad x^t = f^t(A^{t-1}, x^{t-1}), \tag{1}$$

where $f^t$ denotes the transformation that produces new feature map at layer $t$ of the CNN, $A^t$ represents the aggregated information up to $(t-1)$th layer, and $g^t$ is the corresponding transformation that summarizes these layers. For CNNs without skip connections, according to our definition, they do not involve layer aggregation since $x^t = f^t(x^{t-1})$, i.e., the information in $x^l$s with $l < t - 1$ is only available to $x^t$ through $x^{t-1}$. On the other hand, the Hierarchical Layer Aggregation [59] can be shown to have such mechanism since it satisfies Eq. (1).

It is usually more convenient to consider an additive form for the summarizing transformation, i.e.

$$A^t = \sum_{l=0}^{t-1} g_l^t(x^l), \tag{2}$$

where an additive constant $c$ may also be included, and $g_l^t$ can be one or more convolutions, e.g., a set of 1x1-3x3-1x1 convolutions that compose a bottleneck block. Consider a DenseNet layer

$$x^t = \text{Conv3}^t \left[ \text{Conv1}^t(\text{Concat}(x^0, x^1, \ldots, x^{t-1})) \right], \tag{3}$$

where $\text{Conv1}^t$ and $\text{Conv3}^t$ denote 1x1 and 3x3 convolutions, respectively. It can be verified that

$$\text{Conv1}^t(\text{Concat}(x^0, x^1, \ldots, x^{t-1})) = \sum_{l=0}^{t-1} \text{Conv1}_l^t(x^l), \tag{4}$$

where the kernel weights of $\text{Conv1}_l^t$ form a partition of the weights in $\text{Conv1}^t$. As a result,

$$A^t = \sum_{l=0}^{t-1} \text{Conv1}_l^{t+1}(x^l) \quad \text{and} \quad x^t = \text{Conv3}^t \left[ A^{t-1} + \text{Conv1}_{t-1}^t(x^{t-1}) \right],$$

i.e., it is a typical example of the layer aggregation mechanism.

We next consider the pre-activated ResNets [21]. On the one hand, according to the ResNet update $x^t = x^{t-1} + g_{t-1}(x^{t-1})$, it may be concluded to involve no layer aggregation. On the other hand, by recursively applying the update, we have $x^t = \sum_{l=0}^{t-1} g_l(x^l) + x^0$, which satisfies Eq. (1) since $A^t = \sum_{l=0}^{t-1} g_l(x^l) + x^0$ and $x^t = A^{t-1} + g_{t-1}(x^{t-1})$. It is noticeable that $g_l$ depends only on $l$, the ordinal number of previous layers, and we will discuss such a pattern in the next subsection.

**Connection with channel attention** The layer aggregation mechanism in the original DenseNet can be interpreted as channel attention across layers. Specifically, the *multi-head masked self-attention* [50] mechanism can be written as

$$\text{Attention}^t = \text{Concat}(\text{head}_1^t, ..., \text{head}_m^t) \text{ with } \text{head}_i^t = \sum_{l=0}^{t-1} s_{l,i}^t x^l, \text{ for } i = 1, ..., m, \quad (5)$$

where query, key and value are all set to be $x^t$, $s_{l,i}^t$s are scalar similarity measures and $m$ is the number of attention heads.

Note that, when considering DenseNet layers, feature maps are tensors instead of matrices. Suppose that all $x^l$'s have the same number of channels, denoted by $k$, and then $\text{Conv1}^t$ has $tk$ input and $m$ output channels. Denote the $tk \times 1 \times 1 \times m$ kernel weights of $\text{Conv1}^t$ by $\{W_{j,i}^t\}$. Let $\text{head}_i^t$ be the $i$th channel in $\text{Attention}^t$, and $x_c^l$ be the $c$th channel of the feature tensor $x^l$, then it holds that $\text{head}_i^t = \sum_{l,c} W_{j(l,c),i}^t x_c^l$, where $j(l,c) = lk + c$ is an index. As a result, DenseNet updates coincide with a natural extension of Eq. (5) to tensor variables.

However, the above attention mechanism is too meticulous in a CNN context. For example, it will collapse when simply changing $\text{Conv1}^t$ in DenseNets to a 3x3 convolution. In contrast, the layer aggregation interpretation of DenseNet is robust to such a modification.

Our purpose of the discussions above is two-fold: (1) we build the relationship that layer aggregation generalizes channel attention across layers based on the DenseNet architecture; and thus, (2) layer aggregation is not a substitute to channel attention modules, but complementary to them, which inspires us to add the proposed RLA modules to ECAnets in Section 4.

## 3.2 Sequential structure of layers in deep networks

The complexity of summarizing transformations can drop dramatically from (1) to (2), while it may still be complicated. Actually DenseNets have been frequently criticized for their redundancy. Note that the layers $\{x^t\}$ form a sequential structure, and hence this subsection will introduce some related concepts in time series analysis such that we can further compress the summarizing transformations at (1) and (2). More background on time series is available in the supplementary material A.2.

We first simplify Eq. (2) by assuming that $g_l^t$ depends on the lag $t - l$ only, and this is a typical setting in time series analysis. If all $x^t$s are scalars and all related transformations are linear, we have

$$x^t = \beta_1 x^{t-1} + ... + \beta_{t-1} x^1 + \beta_t x^0 = \sum_{l=0}^{t-1} \beta_{t-l} x^l. \quad (6)$$

It can also be rewritten into $\sum_{l=1}^{\infty} \beta_l x^{t-l}$ with $x^s = 0$ for $s < 0$, which has a form of autoregressive (AR) models in time series, $x^t = \sum_{l=1}^{\infty} \beta_l x^{t-l} + i^t$, with $i^t$ being the additive error.

In time series analysis, the above AR model, which is usually referred to as the AR$(\infty)$ model, is mainly for theoretical discussions, and the commonly used running model is the autoregressive moving average (ARMA) model since it has a parsimonious structure. Consider a simple ARMA$(1, 1)$ model, $x^t = \beta x^{t-1} + i^t - \gamma i^{t-1}$. It can be verified to have an AR$(\infty)$ form of

$$\begin{aligned} x^t &= \beta x^{t-1} + i^t - \gamma i^{t-1} = \beta x^{t-1} - \gamma(x^{t-1} - \beta x^{t-2}) + i^t - \gamma^2 i^{t-2} \\ &= \cdots = \sum_{l=1}^{\infty} (\beta - \gamma)\gamma^{l-1} x^{t-l} + i^t, \end{aligned} \quad (7)$$

where equation $i^s = x^s - \beta x^{s-1} + \gamma i^{s-1}$ is recursively applied for all $s \leq t - 1$, while there are only two parameters involved. This motivates us to use a similar pattern to construct a more parsimonious form for Eq. (6). Let

$$h^t = \alpha x^{t-1} + \gamma h^{t-1} \quad \text{and} \quad x^t = \beta_1 x^{t-1} + \beta_2 h^{t-1}. \quad (8)$$

Assuming $h^0 = 0$, we then have $h^t = \sum_{l=1}^{t} \alpha \gamma^{l-1} x^{t-l}$ and $x^t = \beta_1 x^{t-1} + \beta_2 \alpha \sum_{l=1}^{t-1} \gamma^{l-1} x^{t-1-l}$, which has a form of (6) but involves at most four parameters. It is worthy to point out that $h^t$ plays a role similar to that of $i^t$, and Eq. (8) is similar to the principle of RNNs [11].

We may also consider the case that $g_l^t$ at (2) depends on $l$ only, i.e. $x^t = f^t(\sum_{l=0}^{t-2} g_l(x^l), x^{t-1})$, which leads to a ResNet-type layer. From (8), for $s > 0$, the information in $x^{t-s}$ will be reused when extracting features $x^t$ by an amount that depends on lag $s$ only and decays at a rate of $\gamma^s$. However, as mentioned in Section 3.1, ResNet can be viewed as layer aggregation in a different pattern, and the amount of information in $x^l$ reused when extracting features $x^t$ depends on the ordinal number $l$ only. We next conduct a small experiment to compare the ResNet-type simplification with that at (6).

Table 1: Classification error of DenseNet and its parameter sharing variants on the CIFAR-10 dataset.

| Model | Params | Error (%) |
|---|---|---|
| DenseNet | 0.80M | $5.80\pm 0.33$ |
| - Shared-Lag | 0.60M | 6.22 |
| - Shared-Ordinal | 0.60M | 6.29 |

**Comparison of two simplifications**  The experiment is carried out based on the 100-layer DenseNet-BC ($k = 12$) [28]. According to the two simplifications, we modify the DenseNet update (3) into the following versions:

$$\text{(Shared-Lag)} \quad x^t = \text{Conv3}^t(\text{Conv1}_1(x^{t-1}) + \cdots + \text{Conv1}_{t-1}(x^1) + \text{Conv1}_0^t(x^0)), \quad (9)$$

$$\text{(Shared-Ordinal)} \quad x^t = \text{Conv3}^t(\text{Conv1}_{t-1}(x^{t-1}) + \cdots + \text{Conv1}_1(x^1) + \text{Conv1}_0^t(x^0)), \quad (10)$$

which correspond to (6) and ResNet, respectively. The above formulations require $\text{Conv1}_l$ to be compatible with all previous layers. For DenseNet-BC ($k = 12$), the input to dense layers $x^0$ has $3 + 2 \times k = 27$ channels, while all later layers $x^l, 1 \leq l \leq 16$, have $k = 12$ channels. Thus, we keep the convolutions applied to $x^0$ unchanged and only modify the other 1x1 convolutions. Detailed explanations of the network structure of (9) are provided in the supplementary material A.2.

Table 1 presents the comparison of parameter count and accuracy on CIFAR-10. Detailed training settings are the same as other experiments on CIFAR-10 in Section 4. Sharing the 1x1 convolutions reduces the number of parameters by 25% (0.20M), while the accuracy only drops by about 0.5%. This observation is consistent with the parameter sharing experiment by [37], where they found that a ResNet with shared weights retains most of its performance on CIFAR-10. And the better performance of Shared-Lag encourages us to pursue a more parsimonious structure along the direction of (6).

Furthermore, we extract the weights learned by the shared 1x1 convolutions and produce Figure 2, following Figure 5 in [28]. A first observation is that the strongest signal in Shared-Lag (lag=1) is almost twice as strong as that in Shared-Ordinal (layer=1). Secondly, we spot a quick decaying pattern in the plot for Shared-Lag, which is similar to the *exponential decay* pattern in an ARMA model or an RNN [61]. This observation provides us empirical evidence to adopt the principle at (8) to design a light-weighted layer aggregation mechanism.

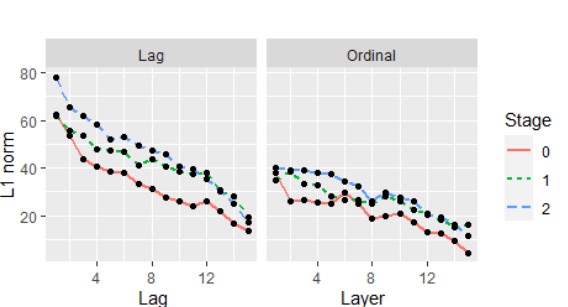

Figure 2: $L_1$ norm of the shared weights.

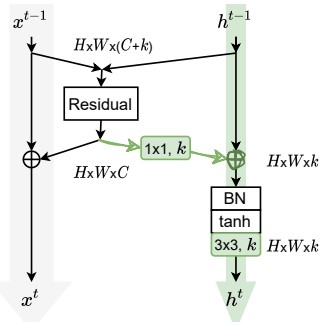

Figure 3: Diagram of an RLA module.

### 3.3  Recurrent layer aggregation

By adopting the idea at (8), we introduce a recurrent layer aggregation (RLA) mechanism below,

$$h^t = g^t(h^{t-1}, x^{t-1}) \quad \text{and} \quad x^t = f^t(h^{t-1}, x^{t-1}), \quad (11)$$

where $A^t = h^t$ represents the recurrently aggregated information up to $(t - 1)$th layer, and it can be shown to have an additive form of (2) under a linearity assumption; see supplementary material A.3. By considering different forms of $g^t$, we can achieve various *RLA modules* to help the main CNN better extract features, and weight sharing can be further used to regularize the layer aggregation. The form of $f^t$ is mainly determined by the currently used mainstream CNNs, and the combination of aggregated information may be implemented in different ways.

Table 2: Classification error (%) on CIFAR-10 and CIFAR-100 test sets. Results are highlighted in bold if RLA-Net outperforms our re-implemented baseline model shown in the column "Re-impl.".

| Dataset | Model | Params | FLOPs | Original | Re-impl. | RLA-Net | $\Delta$ Params | $\Delta$FLOPs |
|---------|-------|--------|-------|----------|----------|---------|---------|---------|
| C-10 | ResNet-110 | 1.73M | 8.67M | 6.37 | 6.35 | **5.88** | +0.07M | +0.37M |
| | ResNet-164 | 1.72M | 8.55M | 5.46 | 5.72 | **4.95** | +0.02M | +0.19M |
| | Xception | 2.70M | 13.54M | - | 6.03 | **5.78** | +0.07M | +0.33M |
| C-100 | ResNet-110 | 1.74M | 8.69M | 26.88 | 28.51 | **27.44** | +0.07M | +0.38M |
| | ResNet-164 | 1.74M | 8.67M | 24.33 | 25.22 | **23.78** | +0.02M | +0.19M |
| | Xception | 2.75M | 13.77M | - | 24.31 | **24.00** | +0.06M | +0.33M |

Figure 3 gives an example for a CNN block with a residual connection, and legends are provided at the bottom of Figure 4. Its update can be formulated as

$$h^t = g_2[g_1(y^t) + h^{t-1}] \quad \text{and} \quad x^t = y^t + x^{t-1},$$

where $y^t = f_1^t[\text{Concat}(h^{t-1}, x^{t-1})]$. Specifically, inputs $x^{t-1}$ and $h^{t-1}$ are combined via concatenation and then passed to a residual unit to extract new features. One copy of this new feature map is joined with the residual connection, and the other copy is compressed via the shared 1x1 convolution and then added to $h^{t-1}$. Then, the RLA path goes through a pre-activated ConvRNN unit comprised of an unshared batch normalization, a tanh activation, and a recurrent 3x3 convolution.

Practically, backbone CNNs often comprise of several stages, and RLA modules can cater to various stage structures following the guidelines below:

1. Group main CNN layers/blocks into stages so that feature maps within the same stage have the same resolution, e.g., ResNet-50/101/152 has 4 stages and MobileNetV2 has 5 stages.

2. For each stage, define (a) a shared recurrent unit, allowing information to be compressed differently at various stages, and (b) a shared 1x1 convolution, to change the channels of the residual feature maps for adding it to the RLA hidden state.

3. (*RLA path*) RLA hidden state $h^0$ is initialized as zero. Before entering the next stage, the RLA hidden states are downsampled to the correct resolution via average pooling.

4. (*RLA output*) For classification, the last hidden state is concatenated with the main CNN output and passed to the classifier. For detection and segmentation, RLA output can be discarded.

## 4 Experiments

We verify the effectiveness of our RLA module in Figure 3 on image classification, object detection and instance segmentation tasks using CIFAR, ImageNet and MS COCO datasets. All experiments are implemented on four Tesla V100 GPUs. [1] More implementation details and tabular descriptions of network architectures are provided in Section B of the supplementary material.

### 4.1 CIFAR and analysis

The two CIFAR datasets, CIFAR-10 and CIFAR-100, consist of 60k $32 \times 32$ RGB images of 10 and 100 classes. The training and test sets contain 50k and 10k images. We adopt a train/validation split of 45k/5k and follow the widely used data preprocessing scheme in [20, 28]. All the networks are trained from scratch using Nesterov SGD with momentum of 0.9, $l_2$ weight decay of $10^{-4}$, and a batch size of 128 for 300 epochs. The initial learning rate is set to 0.1 and is divided by 10 at 150 and 225 epochs. We choose the model with the highest validation accuracy.

We report the performance of each baseline model and its RLA-Net counterpart on CIFAR-10 and CIFAR-100 test sets in Table 2. We observe that RLA-Net consistently improves performance across various networks with different depths, blocks and patterns of stages. For the three models in Table 2, adding the RLA module introduces about a 1%-4% increase in parameters and FLOPs. Meanwhile,

---

[1]Our implementation and weights are available at `https://github.com/fangyanwen1106/RLANet`.

Table 3: Comparisons of single-crop error on the ILSVRC 2012 validation set with center crop of size $224 \times 224$. All models were trained with a crop size of $224 \times 224$. The last column report increases in Top-1 accuracy by adding RLA. † Trained with some training tricks and architectural refinements in [23]; see Section B2.2.

| Baseline+Module | Params | FLOPs | Top-1 err. | Top-5 err. | ↑Acc1 |
|---|---|---|---|---|---|
| ResNet-50 [19] | 24.37M | 3.83G | 24.70 | 7.80 | |
| +RLA (Ours) | 24.67M | 4.17G | 22.83 | 6.58 | 1.87 |
| +SE [26] | 26.77M | 3.84G | 23.29 | 6.62 | |
| +CBAM [55] | 26.77M | 3.84G | 22.66 | 6.31 | |
| +ECA [54] | 24.37M | 3.83G | 22.52 | 6.32 | |
| +ECA+RLA (Ours) | 24.67M | 4.18G | **22.15** | **6.11** | 0.37 |
| ResNet-101 [19] | 42.49M | 7.30G | 23.60 | 7.10 | |
| +RLA (Ours) | 42.92M | 7.79G | 21.48 | 5.80 | 2.12 |
| +SE [26] | 47.01M | 7.31G | 22.38 | 6.07 | |
| +CBAM [55] | 47.01M | 7.31G | 21.51 | 5.67 | |
| +ECA [54] | 42.49M | 7.30G | 21.35 | 5.66 | |
| +ECA+RLA (Ours) | 42.92M | 7.80G | **21.13** | **5.61** | 0.22 |
| ResNet-152 [19] | 57.40M | 10.77G | 23.00 | 6.70 | |
| +RLA (Ours) | 57.96M | 11.47G | 21.22 | 5.65 | 1.78 |
| +SE [26] | 63.68M | 10.78G | 21.57 | 5.73 | |
| +ECA [54] | 57.41M | 10.78G | 21.08 | **5.45** | |
| +ECA+RLA (Ours) | 57.96M | 11.48G | **20.66** | 5.49 | 0.42 |
| MobileNetV2 [54, 43] | 3.34M | 299.6M | 28.36 | 9.80 | |
| +RLA (Ours) | 3.46M | 351.8M | 27.68 | **9.18** | 0.68 |
| +SE [26] | 3.51M | 300.3M | 27.58 | 9.33 | |
| +ECA [54] | 3.34M | 300.1M | 27.44 | 9.19 | |
| +ECA+RLA (Ours) | 3.46M | 352.4M | **27.07** | **8.89** | 0.37 |
| ResNet-50-D† | 24.37M | 3.83G | 20.57 | 5.31 | |
| +RLA (Ours) | 24.67M | 4.17G | 20.09 | 5.11 | 0.48 |
| +ECA | 24.37M | 3.83G | 19.88 | 4.76 | |
| +ECA+RLA (Ours) | 24.67M | 4.18G | **19.28** | **4.63** | 0.60 |
| ResNet-101-D† | 42.49M | 7.30G | 18.89 | 4.51 | |
| +RLA (Ours) | 42.92M | 7.79G | 18.46 | 4.25 | 0.43 |
| +ECA | 42.49M | 7.30G | 18.34 | 4.01 | |
| +ECA+RLA (Ours) | 42.92M | 7.80G | **18.03** | **3.89** | 0.31 |
| DenseNet-161 ($k = 48$) [54] | 27.35M | 7.34G | 22.35 | 6.20 | |
| DenseNet-264 ($k = 32$) [54] | 31.79M | 5.52G | 22.15 | 6.22 | |
| ResNet-200 [54] | 74.45M | 14.10G | 21.80 | 6.00 | |
| DPN-68 [58] | 12.80M | 2.50G | 23.60 | 6.90 | |
| DPN-92 [58] | 38.00M | 6.50G | 20.70 | 5.40 | |
| DPN-98 [58] | 61.60M | 11.70G | 20.20 | 5.20 | |
| AOGNet-12M [58] | 11.90M | 2.40G | 22.30 | 6.10 | |
| AOGNet-40M [58] | 40.30M | 8.90G | 19.80 | 4.90 | |
| HCGNet-B [58] | 12.90M | 2.00G | 21.50 | 5.90 | |
| HCGNet-C [58] | 42.20M | 7.10G | 19.50 | 4.80 | |

our RLA-Net improves the accuracies of ResNet-110 and ResNet-164 by about 0.5% and 1.0% on CIFAR-10 and CIFAR-100, respectively.

## 4.2 ImageNet classification

This subsection reports experiments on the ImageNet LSVRC 2012 dataset [12]. We employ our RLA module on the widely used ResNet [20] architectures and the light-weighted MobileNetV2 [43], as well as their counterparts with the state-of-the-art channel attention ECA modules [54]. For MobileNetV2, depthwise separable convolutions are used for the shared 3x3 convolutions to adapt to the inverted residual blocks of MobileNetV2. When both RLA and ECA modules present, ECA modules are placed according to [54], and the output of the ECA module is then passed to our RLA

Table 4: Object detection results of different methods on COCO val2017.

| Methods | Detector | $AP$ | $AP_{50}$ | $AP_{75}$ | $AP_S$ | $AP_M$ | $AP_L$ | $\uparrow AP$ |
|---|---|---|---|---|---|---|---|---|
| ResNet-50 | | 36.4 | 58.2 | 39.2 | 21.8 | 40.0 | 46.2 | |
| +RLA (Ours) | | **38.8** | 59.6 | **42.0** | 22.5 | **42.9** | **49.5** | 2.4 |
| +SE | | 37.7 | 60.1 | 40.9 | 22.9 | 41.9 | 48.2 | |
| +ECA | | 38.0 | 60.6 | 40.9 | 23.4 | 42.1 | 48.0 | |
| +ECA+RLA (Ours) | Faster R-CNN | **39.8** | **61.2** | **43.2** | **23.9** | **43.7** | **50.8** | 1.8 |
| ResNet-101 | | 38.7 | 60.6 | 41.9 | 22.7 | 43.2 | 50.4 | |
| +RLA (Ours) | | **41.2** | 61.8 | **44.9** | 23.7 | **45.7** | **53.8** | 2.5 |
| +SE | | 39.6 | 62.0 | 43.1 | 23.7 | 44.0 | 51.4 | |
| +ECA | | 40.3 | 62.9 | 44.0 | 24.5 | 44.7 | 51.3 | |
| +ECA+RLA (Ours) | | **42.1** | **63.3** | **46.1** | **24.9** | **46.4** | **54.8** | 1.8 |
| ResNet-50 | | 35.6 | 55.5 | 38.2 | 20.0 | 39.6 | 46.8 | |
| +RLA (Ours) | | **37.9** | 57.0 | **40.8** | **22.0** | **41.7** | 49.2 | 2.3 |
| +SE | | 37.1 | 57.2 | 39.9 | 21.2 | 40.7 | 49.3 | |
| +ECA | | 37.3 | 57.7 | 39.6 | 21.9 | 41.3 | 48.9 | |
| +ECA+RLA (Ours) | RetinaNet | **38.9** | **58.7** | **41.7** | **23.9** | **42.7** | **49.7** | 1.6 |
| ResNet-101 | | 37.7 | 57.5 | 40.4 | 21.1 | 42.2 | 49.5 | |
| +RLA (Ours) | | **40.3** | 59.8 | **43.5** | **24.2** | **43.8** | **52.7** | 2.6 |
| +SE | | 38.7 | 59.1 | 41.6 | 22.1 | 43.1 | 50.9 | |
| +ECA | | 39.1 | 59.9 | 41.8 | 22.8 | 43.4 | 50.6 | |
| +ECA+RLA (Ours) | | **41.5** | **61.6** | **44.4** | **25.3** | **45.7** | **53.8** | 2.4 |

module through the shared 1x1 convolution. We compare our RLA-Nets with several state-of-the-art attention-based CNN architectures, including SENet [26], CBAM [55] and ECANet [54].

For training RLA-ResNets, we follow the same data augmentation and hyper-parameter settings as in [20, 28]. All the networks are trained using SGD with momentum 0.9, $l_2$ weight decay of $10^{-4}$, and a mini-batch size of 256 on 4 GPUs. We train models for 120 epochs from scratch and use the weight initialization strategy described in [18]. The initial learning rate is set to 0.1 and decreased by a factor of 10 every 30 epochs. For the light-weighted model MobileNetV2, we train the model on 2 GPUs within 400 epochs using SGD with weight decay of 4e-5, momentum of 0.9, and a mini-batch size of 96, following the settings in [43]. The initial learning rate is set to 0.045, decreased by a linear decay rate of 0.98. All the parameters and FLOPs shown in the tables are computed by our devices.

Table 3 shows that our RLA-Nets obtain better results compared with their original counterparts while introducing small extra costs. Though RLA modules introduce more FLOPs when compared with channel attention modules, we find that the extra training time per epoch is almost the same. Specifically, adding SE, ECA or RLA module to ResNet-101 costs about 15%, 15% or 19% more training time. RLA-ResNets introduce about 1% increase in the number of parameters, leading to 1.9%, 2.1% and 1.8% increases in top-1 accuracy for RLA-ResNet-50/101/152, respectively. Similar results can be observed on the light-weighted MobileNetV2 architectures. Furthermore, combinations of RLA and ECA modules yield remarkably better results, achieving the best performances among these models. These further improvements based on ECANets verify our perspective that the functionalities of channel attention modules and our proposed RLA module are complementary.

Last but not least, our model with some tricks in [23] can perform remarkably better than many popular networks and state-of-the-art networks with delicate designs, including DPN [7], AOGNet [35] and HCGNet [58], which are designed to combine the advantages of ResNet and DenseNet. The notable and consistent improvements demonstrate that any tricks or refinements used to further improve the original ResNets can be directly applied to our model to further improve the model performance. And RLA modules can be treated as one of the beneficial architectural refinements to be adopted by state-of-the-art CNNs as well.

## 4.3 Object detection and instance segmentation on MS COCO

To show the transferability and the generalization ability, we experiment our RLA-Net on object detection task using Faster R-CNN [42], Mask R-CNN [22] and RetinaNet [39] as detectors. For Mask R-CNN, we also show instance segmentation results. All detectors are implemented by the open

Table 5: Object detection and instance segmentation results of different methods using Mask R-CNN on COCO val2017. $AP^{bb}$ and $AP^m$ denote AP of bounding box detection and instance segmentation.

| Methods | $AP^{bb}$ | $AP^{bb}_{50}$ | $AP^{bb}_{75}$ | $AP^m$ | $AP^m_{50}$ | $AP^m_{75}$ | $\uparrow AP^{bb}$ | $\uparrow AP^m$ |
|---|---|---|---|---|---|---|---|---|
| ResNet-50 | 37.2 | 58.9 | 40.3 | 34.1 | 55.5 | 36.2 | | |
| +RLA (Ours) | **39.5** | 60.1 | **43.3** | **35.6** | 56.9 | **38.0** | 2.3 | 1.5 |
| +SE | 38.7 | 60.9 | 42.1 | 35.4 | 57.4 | 37.8 | | |
| +1 NL | 38.0 | 59.8 | 41.0 | 34.7 | 56.7 | 36.6 | | |
| +GC Block | 39.4 | 61.6 | 42.4 | 35.7 | 58.4 | 37.6 | | |
| +ECA | 39.0 | 61.3 | 42.1 | 35.6 | 57.1 | 37.7 | | |
| +ECA+RLA (Ours) | **40.6** | **61.8** | **44.0** | **36.5** | **58.4** | **38.8** | 1.6 | 0.9 |
| ResNet-101 | 39.4 | 60.9 | 43.3 | 35.9 | 57.7 | 38.4 | | |
| +RLA (Ours) | **41.8** | 62.3 | **46.2** | 37.3 | 59.2 | **40.1** | 2.4 | 1.4 |
| +SE | 40.7 | 62.5 | 44.3 | 36.8 | 59.3 | 39.2 | | |
| +ECA | 41.3 | 63.1 | 44.8 | 37.4 | 59.9 | 39.8 | | |
| +ECA+RLA (Ours) | **42.9** | **63.6** | **46.9** | **38.1** | **60.5** | **40.8** | 1.6 | 0.7 |

source MMDetection toolkit [4]. We employ the same settings as in [54] to finetune our RLA-Nets on COCO train2017 set. Specifically, the shorter side of input images are resized to 800. We train all detectors within 12 epochs using SGD with weight decay of 1e-4, momentum of 0.9 and mini-batch size of 8. The learning rate is initialized to 0.01 and is decreased by a factor of 10 after 8 and 11 epochs, respectively, i.e., the 1x training schedule. Different from the classification task, we do not pass the hidden states of our RLA module to the FPN. Thus, the gains are solely from the more powerful representations learnt in the main CNNs.

We report the results on COCO val2017 set by standard COCO metrics of Average Precision (AP). Tables 4 and 5 report the performances of the object detectors. We can observe that our RLA-Net outperforms the original ResNet by 2.4% and 2.5% in terms of AP for the networks of 50 and 101 layers, respectively. More excitingly, our RLA-ResNet-50 outperforms ResNet-101 on these three detectors. In particular, our RLA module achieves more gains in $AP_{75}$ and $AP_L$, indicating the high accuracy and effectiveness for large objects. Remarkably, exploiting RLA on ECANets can surpass all other models with large margins. We hypothesize that, with the help of RLA modules, positional information from low-level features is better preserved in the main CNN, leading to these notable improvements. In summary, the results in Tables 4 and 5 demonstrate that our RLA-Net can well generalize to various tasks with extraordinary benefits on the object detection task.

## 4.4 Ablation study

We conduct an ablation study on ImageNet with ResNet-50 as the baseline network. To validate the effectiveness of the detailed inner structure, we ablate the important design elements of RLA module and compare the proposed structure with the following variants:

(a) channels increased by 32;

(b) RLA without parameter sharing;

(c) RLA without the information flow from RLA to the main CNN (as in [41]);

(d) RLA with ConvLSTM cell;

(e) RLA with the post-activated hidden unit; and

(f-j) RLA with different connectivity between two paths, see Figure 4.

The results are reported in Table 6. We first demonstrate that the improvement in accuracy is not fully caused by the increment in network width by comparing to the variant with 32 additional channels in each layer. We then investigate the necessity of recurrence in our module by comparing it with its fully unshared counterpart. Results show that the recurrent design can reduce parameter count and slightly increase accuracy at the same time. Removing the information flow from RLA to the main CNN results in a decrease in model performances, validating that the information exchange between RLA and the main CNN is a must. Furthermore, using ConvLSTM cells may be unnecessary since

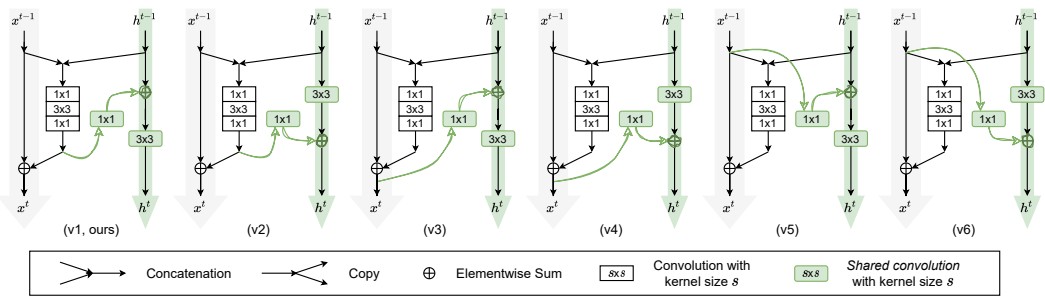

Figure 4: RLA modules with different types of connections between two paths.

Table 6: Classification errors on the ImageNet validation set using ResNet-50 as the baseline model.

| Model | Params (M) | FLOPs (G) | Top-1 err. | Top-5 err. |
|---|---|---|---|---|
| ResNet-50 | 24.37 | 3.83 | 24.70 | 7.80 |
| (a) channel +32 | 25.55 | 4.24 | 23.54 | 6.72 |
| (b) RLA-v1 ($k = 32$, unshared) | 25.12 | 4.17 | 22.93 | 6.61 |
| (c) RLA-v1 ($k = 32$, no exchange) | 24.56 | 4.11 | 23.42 | 6.90 |
| (d) RLA-v1 ($k = 32$, ConvLSTM) | 24.92 | 5.00 | 22.92 | 6.53 |
| (e) RLA-v1 ($k = 32$, PostAct.) | 24.67 | 4.17 | 23.11 | 6.65 |
| Proposed RLA-v1 ($k = 32$) | 24.67 | 4.17 | 22.83 | 6.58 |
| (f) RLA-v2 ($k = 32$) | 24.67 | 4.17 | 23.24 | 6.61 |
| (g) RLA-v3 ($k = 32$) | 24.67 | 4.17 | 22.95 | 6.54 |
| (h) RLA-v4 ($k = 32$) | 24.67 | 4.17 | 23.36 | 6.72 |
| (i) RLA-v5 ($k = 32$) | 24.67 | 4.17 | 23.30 | 6.60 |
| (j) RLA-v6 ($k = 32$) | 24.67 | 4.17 | 23.50 | 6.82 |

each stage only has 3 to 6 time steps in ResNet-50, and the post-activated recurrent unit is not very compatible with the pre-activated network.

We also compare different integration strategies to combine our RLA module with the main CNN. In addition to the proposed design, we consider five variants (v2-v6), as depicted in Figure 4. The performances of the variants are reported in Table 6. We observe that the variants v1, v3 and v5 perform similarly well, as they all integrate new information from the main CNNs before entering into the recurrent operation. Compared with variants v3 and v5, v1 obtains the learned residual as new information, including less redundancy. The performance of our RLA modules is relatively robust to the integration strategies, as long as new information is added before the recurrent operation.

In summary, the proposed RLA module structure in Figure 3 achieves the smallest top-1 error and very competitive top-5 error with the minimum number of additional parameters. This structure can strike a good balance between the increase of computational cost and performance gain. We also conduct a more comprehensive ablation study on the CIFAR-10 dataset with ResNet-164 as the baseline network. Due to limited space, we present details of the ablation studies and the investigated module structures in the supplementary material B.4.

## 5   Conclusion and future work

This paper proposes a recurrent layer aggregation module that is light-weighted and capable of providing the backbone CNN with sequentially aggregated information from previous layers. Its potential has been well illustrated by the applications on mainstream CNNs with benchmark datasets. On the other hand, we mainly focused on controlled experiments to illustrate the effectiveness and compatibility of the RLA modules with settings adopted from existing work. Our first future work is to consider hyperparameter tuning, advanced training tricks [23, 2], or architecture search [49]. Moreover, in terms of RLA design and application, it is also an interesting direction to apply it at the stage level on top of the layer level, which leads to deep RLA modules.

## Acknowledgments and Disclosure of Funding

We thank the anonymous reviewers for their constructive feedback and Jinxu Zhao for technical support. This project is supported by National Supercomputer Center in Guangzhou, China.

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
