# Supplementary Materials for
# Recurrence along Depth: Deep Convolutional Neural Networks with Recurrent Layer Aggregation

**Jingyu Zhao, Yanwen Fang and Guodong Li**
Department of Statistics and Actuarial Science
The University of Hong Kong
{gladys17, u3545683}@connect.hku.hk, gdli@hku.hk

## Abstract

This document contains supplementary explanations and experiments to support the proposed recurrent layer aggregation (RLA) module. Section A supplements the arguments made in Section 3. Section B provides more experiment details and results. For easy reference, subsections are ordered the same as those in the main paper, and we may also repeat some necessary contents here.

## A    Recurrent Layer Aggregation Modules

### A.1    Layer Aggregation

This subsection provides more explanations of the layer aggregation mechanism. In the main paper, Eq. (1) gives the general definition

$$A^t = g^t(x^{t-1}, x^{t-2}, \dots, x^0) \quad \text{and} \quad x^t = f^t(A^{t-1}, x^{t-1}),$$

and Eq. (2) considers the simplified additive form

$$A^t = \sum_{l=0}^{t-1} g_l^t(x^l),$$

which is often satisfied if we impose a linearity assumption and consider the corresponding network without nonlinear activation functions. In the following, we revisit the examples in Section 3.1.

**Example 1. Sequential CNNs without skip connections.**    Such CNNs do not fulfill Eq. (1) or (2) because the information in $x^l$ with $l < t-1$ is only available to $x^t$ through $x^{t-1}$. More rigorously, we may write the features at the $t$-th layer as

$$x^t = \text{Conv}_t(x^{t-1}),$$
$$x^t = \text{Conv}_t(\text{Conv}_{t-1}(x^{t-2})),$$
$$\vdots$$
$$x^t = \text{Conv}_t(\text{Conv}_{t-1}(\cdots \text{Conv}_1(x^0))),$$

which do not have the form of Eq. (1) or (2). Here, $\text{Conv}_l(\cdot)$ for $1 \le l \le t$, denotes convolution operations at the $l$th layer.

**Example 2. Hierarchical Layer Aggregation (HLA) [16].**    HLA satisfies Eq. (1) since its base aggregation operation $N$ (see Eq. 3 in [16]) is given by

$$N(x^1, ..., x^n) = \sigma(\text{BatchNorm}(\sum_i W_i x^i + b)).$$

35th Conference on Neural Information Processing Systems (NeurIPS 2021).

Furthermore, if we ignore nonlinearity $\sigma$ and batch normalization and rewrite weights multiplication as a convolution function $\text{Conv}_i(\cdot)$, we have

$$N(x^1, ..., x^n) = \sum_i \text{Conv}_i(x^i),$$

which takes the form of Eq. (2).

**Example 3. DenseNets.** According to [2], Eq. (4)

$$\text{Conv1}^t(\text{Concat}(x^0, x^1, ..., x^{t-1})) = \sum_{l=0}^{t-1} \text{Conv1}_l^t(x^l)$$

holds when dense connectivity is followed by a 1x1 convolution. In Section 3.1, we argued that DenseNets implement channel attention across layers in the sense of [14]. For a DenseNet with growth rate parameter $k$, the corresponding multi-head attention mechanism in Eq. (5) has $m = 4k$ heads, where each head is a weighted average of all channels in previous layers. This interpretation is consistent with the design of attention mechanisms, while a noticeable difference is that the similarity measures are learned in the convolution $\text{Conv1}^t$.

Moreover, changing the $\text{Conv1}^t$ in DenseNets to a convolution with kernel size other than 1x1 will render the weighted average formulation impossible and invalidate its attention interpretation. In the meanwhile, we claimed that such modifications preserve the layer aggregation mechanism. Consider a modified DenseNet layer

$$x^t = \text{Conv3}^t \left[ \text{Conv}^t(\text{Concat}(x^0, x^1, ..., x^{t-1})) \right],$$

which can still be rewritten as

$$A^t = \sum_{l=0}^{t-1} \text{Conv}_l^{t+1}(x^l) \quad \text{and} \quad x^t = \text{Conv3}^t \left[ A^{t-1} + \text{Conv}_{t-1}^t(x^{t-1}) \right].$$

Thus, the modified DenseNet layer still implements the layer aggregation mechanism.

## A.2 Sequential structure of layers in deep networks

**Time Series Background** In the literature of time series analysis, AR and ARMA models are most commonly used to fit and to predict time series data. For example, given the historical traffic volume $z_1, z_2, ..., z_{t-1}$ on a specific road, the government wants to predict future traffic $z_t, z_{t+1}, ....$ Generally speaking, each observation $z_t$ can depend on all previous ones in a nonlinear manner and the relationship may vary as $t$ differs. AR and ARMA models are linear and assume a formulation where coefficients only depend on the *time lag* between the target and the past observation, shared across different target timestamp $t$. Thus, they are simple, parsimonious and were proved to be useful in many applications.

Time series is a sequence of random variables and the white noise sequence plays a necessary role in time series models as the source of randomness. In the main text, we refer to the white noise term as the additive error because it is an addend in the model equation.

Formally, a $p$th order autoregressive model, a.k.a., an AR($p$) model, $\{Z_t\}$ satisfies

$$Z_t = \theta_0 + \phi_1 Z_{t-1} + \phi_2 Z_{t-2} + \cdots + \phi_p Z_{t-p} + a_t,$$

where $p \geq 0$ is an integer, $\phi$'s are real parameters and $\{a_t\}$ is a white noise sequence. An autoregressive-moving-average model of orders $p$ and $q$, a.k.a., an ARMA($p, q$) model, satisfies

$$Z_t = \theta_0 + \phi_1 Z_{t-1} + \phi_2 Z_{t-2} + \cdots + \phi_p Z_{t-p} + a_t - \theta_1 a_{t-1} - \theta_2 a_{t-1} - \cdots - \theta_q a_{t-q},$$

where $p, q \geq 0$ are integers, $\theta$'s are real parameters for the MA part. *Invertibility* is a property of time series models to characterize whether the information sequence $\{a_t\}$ can be recovered from past observations, and it is always required in time series applications. If an ARMA model is invertible, then it has an *AR representation*

$$Z_t = \pi_1 Z_{t-1} + \pi_2 Z_{t-2} + \cdots + a_t,$$

which takes the form of an AR($\infty$) model, where the infinitely many $\pi$'s can be fully determined by $p + q$ parameters, i.e., $\phi$'s and $\theta$'s. In the main text, we claim that ARMA model simplifies AR and give an example using ARMA($1, 1$); see Eq. (7). We are referring to the fact that any invertible ARMA model has an AR representation, whose parameters can be fully determined by the (much fewer) parameters of the ARMA model.

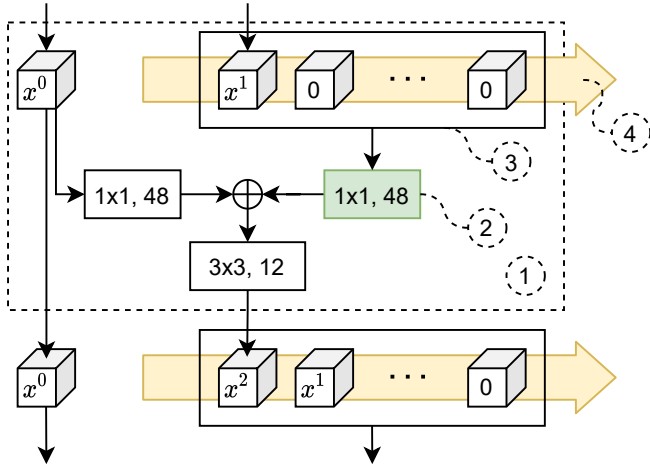

Figure A.1: Visualization of a layer in the Shared-Lag modification of DenseNet in Eq. (9).

**Implementation of two simplifications**   Figure A.1 visualizes the structure of a modified DenseNet layer following Eq. (9), where cubes represent feature tensors, rectangles are convolutions annotated with kernel size and output channels, and $\oplus$ denotes elementwise addition. Numbers inside dashed circles correspond to the following notes:

1. Operations inside the large dashed rectangle represents the 2nd layer in a dense block.
2. This 1x1 convolution is shared among layers within a stage.
3. Dense connections are implemented by this storage area of past layers' output, and 0 is used as a placeholder.
4. This arrow represents a virtual "conveyor belt" indicating that the feature maps are always ordered from the most recent layer to the most distant layer.

$\text{Conv3}^t$ and $\text{Conv1}_0^t(\cdot)$ in Eq. (9) are represented by the unshared 3x3 and 1x1 convolutions (white rectangles), and a combination of $\text{Conv1}_1, \text{Conv1}_2, ..., \text{Conv1}_{15}$ is represented by the shared 1x1 convolution (colored rectangle with note 2). Feature maps from previous layers are ordered such that Eq. (9) holds. The Shared-Ordinal variant of DenseNet can be similarly implemented by adjusting the ordering (see note 3 above) and disabling the "conveyor belt" (see note 4 above).

**Weights of the shared 1x1 convolutions**   Figure 2 in the main paper shows the $L_1$ norm of the weights of the shared 1x1 convolutions. And we observe the following main difference between the two modes of weight sharing:

- When weights are shared based on lag, *the most recent layer* corresponds to the largest weight, and the $L_1$ norm decays as lag increases.
- When weights are shared according to the ordinal number of the layers, *layers closer to the input* correspond to larger weights, and the first few layers have similar weights.

We also claimed that the quick decaying pattern in the plot for Shared-Lag could be exponential decay, and we provide numerical support below. We fit exponential functions to the curves in Figure 2 and use the $R$-squared metric to compare goodness-of-fit. $R^2 \in (0, 1)$ is a metric that can be interpreted as the proportion of variance explained by a model, i.e., a fitted curve. $R^2$ values for the three curves in the Share-Lag subplot are $0.95, 0.86, 0.90$, while that for Shared-Ordinal are $0.77, 0.91, 0.87$, implying poorer fit.

## A.3   Recurrent Layer Aggregation

This subsection shows that the RLA module in Eq. (11) implements layer aggregation as in Eq. (1). And under a linearity assumption, Eq. (11) can also be shown to have the additive form in Eq. (2).

Recall the formulation of RLA mechanism in Eq. (11),

$$h^t = g^t(h^{t-1}, x^{t-1}) \quad \text{and} \quad x^t = f^t(h^{t-1}, x^{t-1}).$$

Recursively substitute $h^s = g^s(h^{s-1}, x^{s-1})$ for $s = t-1, t-2, ...$ into the first equation, we have

$$\begin{aligned}
h^t &= g^t(h^{t-1}, x^{t-1}) \\
&= g^t(g^{t-1}(h^{t-2}, x^{t-2}), x^{t-1}) \\
&= \cdots,
\end{aligned}$$

which is a function of $x^{t-1}, x^{t-2}, ..., x^0$ and a constant $h^0$. Thus, Eq. (1) is satisfied.

If we further assume that there exist functions $g_1^s$ and $g_2^s$ such that $g^s$ and $g_1^s$ satisfy $g^s(h^{s-1}, x^{s-1}) = g_1^s(h^{s-1}) + g_2^s(x^{s-1})$ and $g_1^s(u+v) = g_1^s(u) + g_1^s(v)$ for all $s$, we have

$$\begin{aligned}
h^t &= g_1^t(h^{t-1}) + g_2^t(x^{t-1}) \\
&= g_1^t(g_1^{t-1}(h^{t-2}) + g_2^{t-1}(x^{t-2})) + g_2^t(x^{t-1}) \\
&= g_1^t(g_1^{t-1}(h^{t-2})) + g_1^t(g_2^{t-1}(x^{t-2})) + g_2^t(x^{t-1}) \\
&= \cdots,
\end{aligned}$$

which is a summation over transformed $x^{t-1}, x^{t-2}, ..., x^0$ with a constant $g_1^t(g_1^{t-1}(\cdots g_1^1(h^0)))$.

**The proposed RLA module in Figure 3**  Consider the RLA module given by Figure 3 with update

$$h^t = g_2[g_1(y^t) + h^{t-1}] \quad \text{and} \quad x^t = y^t + x^{t-1},$$

where $y^t = f_1^t[\text{Concat}(h^{t-1}, x^{t-1})]$ and $g_1, g_2$ are the shared 1x1 and 3x3 convolutions. As discussed in the main text, ResNets have a layer aggregation interpretation, i.e., $x^t$ is an aggregation of the residual information $y^t$. According to the ablation study in Sections 4.4 and B.4, it is preferable for the RLA module to perform an aggregation of the residual information $y^t$ instead of the already aggregated $x^t$. Thus, in the following, we show that $h^t$ is an aggregation of $y^t$ instead of $x^t$.

When nonlinearities are ignored, the shared convolution $g_2$ can be distributed to the two terms, i.e.,

$$h^t = g_2(h^{t-1}) + g_2 \circ g_1(y^t),$$

Furthermore, recursively applying the above equation, we have

$$\begin{aligned}
h^t &= g_2(h^{t-1}) + g_2 \circ g_1(y^t) \\
&= g_2(g_2(h^{t-2}) + g_2 \circ g_1(y^{t-1})) + g_2 \circ g_1(y^t) \\
&= g_2^2(h^{t-2}) + g_2^2 \circ g_1(y^{t-1}) + g_2 \circ g_1(y^t) \\
&\vdots \\
&= \sum_{k=1}^{t} g_2^k \circ g_1(y^{t-k+1}) + g_2^t(h^0),
\end{aligned} \tag{A.1}$$

where $\circ$ denotes the composition of convolution functions, and with a slight abuse of notation, the composition of the same function $k$ times is denoted as its $k$-th power, e.g.,

$$g_2^k = \underbrace{g_2 \circ g_2 \circ \cdots \circ g_2}_{k \text{ times}},$$

not to be confused with time varying functions where the superscript denotes the time index. Thus, the RLA hidden feature maps $h^t$ are aggregations of previous residual information.

Moreover, the patterns of layer aggregation in ResNets and the RLA module in Eq. (A.1) are very similar to the AR($\infty$) and ARMA(1, 1) models introduced in Sections 3.2 and A.2. Because for the proposed RLA module in Figure 3, the convolutions $g_l^t$ in Eq. (2) are solely determined by the two shared convolutions $g_2$ and $g_1$.

Table B.1: Network architectures of ResNet-110 (left) and RLA-ResNet-110 (right). Note that each "conv" layer shown in the table corresponds to the sequence BN-ReLU-Conv, except that the first "conv" layer corresponds to Conv-BN-ReLU. RLA channel is specified by the value of k.

| Output size | ResNet-110 | RLA-ResNet-110 |
|---|---|---|
| $32 \times 32$ | conv, $3 \times 3$, 16 | |
| | $\begin{bmatrix} \text{conv}, 3 \times 3, 16 \\ \text{conv}, 3 \times 3, 16 \end{bmatrix} \times 18$ | $\begin{bmatrix} \text{conv}, 3 \times 3, 16 \\ \text{conv}, 3 \times 3, 16 \\ k = 4 \end{bmatrix} \times 18$ |
| $16 \times 16$ | $\begin{bmatrix} \text{conv}, 3 \times 3, 32 \\ \text{conv}, 3 \times 3, 32 \end{bmatrix} \times 18$ | $\begin{bmatrix} \text{conv}, 3 \times 3, 32 \\ \text{conv}, 3 \times 3, 32 \\ k = 4 \end{bmatrix} \times 18$ |
| $8 \times 8$ | $\begin{bmatrix} \text{conv}, 3 \times 3, 64 \\ \text{conv}, 3 \times 3, 64 \end{bmatrix} \times 18$ | $\begin{bmatrix} \text{conv}, 3 \times 3, 64 \\ \text{conv}, 3 \times 3, 64 \\ k = 4 \end{bmatrix} \times 18$ |
| $1 \times 1$ | global average pool, 10-d or 100-d $fc$, softmax | |

# B  Experiments

Due to limited space in the main paper, we present more experiment results, implementation details and descriptions of network structures in this section. Our implementation and weights are available at `https://github.com/fangyanwen1106/RLANet`.

## B.1  Implementation details for CIFAR

The two CIFAR datasets, CIFAR-10 and CIFAR-100, consist of 60k $32 \times 32$ RGB images of 10 and 100 classes, respectively. The training and test sets contain 50k and 10k images, and we adopt a train/validation split of 45k/5k. We follow a standard data preprocessing scheme that is widely used for CIFAR dataset [4, 8]. Specifically, we normalize the data using the RGB-channel means and standard deviations. For data augmentation, the input image is first zero-padded with 4 pixels on each side, resulting in a $40 \times 40$ image. A random $32 \times 32$ crop is then performed on the image or its horizontal flip.

All the networks are trained from scratch using Nesterov SGD with momentum of 0.9, $l_2$ weight decay of $10^{-4}$, and a batch size of 128 for 300 epochs. The initial learning rate is set to 0.1 and is divided by 10 at 150 and 225 epochs. We adopt the weight initialization method following [3]. The model with the highest validation accuracy is chosen and reported. It is noticeable that we use the same hyperparameters as in [8] except that we split validation data and use a batch size of 128. The original $32 \times 32$ images are used for testing. All programs run on one Tesla V100 GPU with 32GB memory.

**RLA-ResNet-110 for CIFAR**   See Table B.1. In ResNet-110, the bottleneck structure in ResNet-164 is replaced with two 3x3 convolutions. To control the number of parameters, the number of channels in ResNet-110 is one-fourth of that in ResNet-164. Following the same spirit, we apply an RLA module with $k = 4$ (instead of 12) for ResNet-110.

**RLA-ResNet-164 for CIFAR**   See Table B.2.

**RLA-Xception for CIFAR**   Since Xception is not originally implemented for the CIFAR dataset, we modify it into a thinner version with 2.7M parameters by reducing the channels in each layer while keeping the depth unchanged. We replace the first 7x7 convolution with a 3x3 one and remove three downsampling layers. We use average pooling instead of max pooling for downsampling. For Xception, the architectures have multiple stages with fewer layers in each stage. We group the layers with the same resolutions as a stage like in ResNet. See Table B.3.

## B.2  ImageNet Classification

Table B.2: Network architectures of ResNet-164 (left) and RLA-ResNet-164 (right). Note that each "conv" layer shown in the table corresponds to the sequence BN-ReLU-Conv, except that the first "conv" layer corresponds to Conv-BN-ReLU. RLA channel is specified by the value of k.

| Output size | ResNet-164 | RLA-ResNet-164 |
|---|---|---|
| $32 \times 32$ | conv, $3 \times 3$, 16 | |
| $32 \times 32$ | $\begin{bmatrix} \text{conv}, 1 \times 1, 16 \\ \text{conv}, 3 \times 3, 16 \\ \text{conv}, 1 \times 1, 64 \end{bmatrix} \times 18$ | $\begin{bmatrix} \text{conv}, 1 \times 1, 16 \\ \text{conv}, 3 \times 3, 16 \\ \text{conv}, 1 \times 1, 64 \\ k = 12 \end{bmatrix} \times 18$ |
| $16 \times 16$ | $\begin{bmatrix} \text{conv}, 1 \times 1, 32 \\ \text{conv}, 3 \times 3, 32 \\ \text{conv}, 1 \times 1, 128 \end{bmatrix} \times 18$ | $\begin{bmatrix} \text{conv}, 1 \times 1, 32 \\ \text{conv}, 3 \times 3, 32 \\ \text{conv}, 1 \times 1, 128 \\ k = 12 \end{bmatrix} \times 18$ |
| $8 \times 8$ | $\begin{bmatrix} \text{conv}, 1 \times 1, 64 \\ \text{conv}, 3 \times 3, 64 \\ \text{conv}, 1 \times 1, 256 \end{bmatrix} \times 18$ | $\begin{bmatrix} \text{conv}, 1 \times 1, 64 \\ \text{conv}, 3 \times 3, 64 \\ \text{conv}, 1 \times 1, 256 \\ k = 12 \end{bmatrix} \times 18$ |
| $1 \times 1$ | global average pool, 10-d or 100-d $fc$, softmax | |

### B.2.1 Implementation details for ImageNet

For training RLA-ResNets, we follow the same data augmentation and hyper-parameter settings as in [4] and [8], which are standard pipelines. Specifically, we apply scale augmentation to the original images. A $224 \times 224$ crop is randomly sampled from a scaled image or its horizontal flip. Each input image is normalized by RGB-channel means and standard deviations. All the networks are trained using SGD with momentum of 0.9, $l_2$ weight decay of $10^{-4}$ and a mini-batch size of 256 on 4x V100 GPUs. We train models for 120 epochs from scratch, and use the weight initialization strategy described in [3]. The initial learning rate is set to 0.1 and decreased by a factor of 10 every 30 epochs. For the light-weighted model MobileNetV2, we trained the model on 2x V100 GPUs within 400 epochs using SGD with weight decay of 4e-5, momentum of 0.9 and mini-batch size of 96, following the settings in [12]. The initial learning rate is set to 0.045, decreased by a linear decay rate of 0.98. For evaluation on the validation set, the shorter side of an input image is first resized to 256, and a center crop of $224 \times 224$ is then used for evaluation.

**RLA-ResNets for ImageNet**   Table B.4 shows the architectures of RLA-ResNet-50, RLA-ResNet-101 and RLA-ResNet-152. According to the experience on CIFAR, we follow the growth rate setting of DenseNet on ImageNet and set $k = 32$ in our RLA module on ImageNet.

**RLA-MobileNetV2 for ImageNet**   We follow the guidelines in Section 3.3 in the main paper, and group main CNN layers of MobileNetV2 into 5 stages. For each stage, a shared 3x3 depthwise separable convolution is applied in the RLA module. Different from ResNets, MobileNetV2 uses inverted residual bottleneck block. To avoid the expansion in model complexity (FLOPs), we make the following modifications according to the properties of the inverted residual block:

1. Instead of concatenating the hidden states $h_{t-1}$ and the inputs $x_{t-1}$ before the first 1x1 Conv in the bottleneck block, $h_{t-1}$ and $x_{t-1}$ are concatenated after the first 1x1 Conv to avoid the expansion step in the inverted residual block.

2. The shared standard convolution is replaced with a shared depthwise separable convolution in the RLA module, which is more compatible with MobileNetV2.

### B.2.2 Experiments with training and architecture refinements

**Implementation details**   In Sections 4.2 and B.2.1, we adopt the standard training pipeline for fair comparisons and validate the effectiveness of the proposed structure. To further improve accuracy, we apply some of training tricks and architecture refinements described in [7]. We apply these training tricks by pytorch-image-models toolkit [15]. [1] Specifically,

---
[1]License: Apache License 2.0

Table B.3: Network architectures of Xception (left) and RLA-Xception (right) used on CIFAR. Note that all "conv" and "sep conv" (depthwise separable convolution) layers are followed by batch normalization. RLA channel is specified by the value of k.

| Output size | Xception | RLA-Xception |
|---|---|---|
| $32 \times 32$ | conv, $3 \times 3$, 16, relu | |
| | conv, $3 \times 3$, 32, relu | |
| | $\begin{bmatrix} \text{sep conv}, 3 \times 3, 64 \\ \text{relu} \\ \text{sep conv}, 3 \times 3, 64 \end{bmatrix}$ | $\begin{bmatrix} \text{sep conv}, 3 \times 3, 64 \\ \text{relu} \\ \text{sep conv}, 3 \times 3, 64 \\ k = 12 \end{bmatrix}$ |
| | $\begin{bmatrix} \text{relu} \\ \text{sep conv}, 3 \times 3, 128 \\ \text{relu} \\ \text{sep conv}, 3 \times 3, 128 \end{bmatrix}$ | $\begin{bmatrix} \text{relu} \\ \text{sep conv}, 3 \times 3, 128 \\ \text{relu} \\ \text{sep conv}, 3 \times 3, 128 \\ k = 12 \end{bmatrix}$ |
| | $\begin{bmatrix} \text{relu} \\ \text{sep conv}, 3 \times 3, 256 \\ \text{relu} \\ \text{sep conv}, 3 \times 3, 256 \end{bmatrix}$ | $\begin{bmatrix} \text{relu} \\ \text{sep conv}, 3 \times 3, 256 \\ \text{relu} \\ \text{sep conv}, 3 \times 3, 256 \\ k = 12 \end{bmatrix}$ |
| $16 \times 16$ | $\begin{bmatrix} \text{relu} \\ \text{sep conv}, 3 \times 3, 256 \\ \text{relu} \\ \text{sep conv}, 3 \times 3, 256 \\ \text{relu} \\ \text{sep conv}, 3 \times 3, 256 \end{bmatrix} \times 8$ | $\begin{bmatrix} \text{relu} \\ \text{sep conv}, 3 \times 3, 256 \\ \text{relu} \\ \text{sep conv}, 3 \times 3, 256 \\ \text{relu} \\ \text{sep conv}, 3 \times 3, 256 \\ k = 12 \end{bmatrix} \times 8$ |
| | $\begin{bmatrix} \text{relu} \\ \text{sep conv}, 3 \times 3, 256 \\ \text{relu} \\ \text{sep conv}, 3 \times 3, 512 \end{bmatrix}$ | $\begin{bmatrix} \text{relu} \\ \text{sep conv}, 3 \times 3, 256 \\ \text{relu} \\ \text{sep conv}, 3 \times 3, 512 \\ k = 12 \end{bmatrix}$ |
| $8 \times 8$ | $\begin{bmatrix} \text{sep conv}, 3 \times 3, 512 \\ \text{relu} \\ \text{sep conv}, 3 \times 3, 512 \\ \text{relu} \end{bmatrix}$ | $\begin{bmatrix} \text{sep conv}, 3 \times 3, 512 \\ \text{relu} \\ \text{sep conv}, 3 \times 3, 512 \\ \text{relu} \\ k = 12 \end{bmatrix}$ |
| $1 \times 1$ | global average pool, 10-d or 100-d $fc$, softmax | |

(a) we apply label smoothing with $\epsilon = 0.1$ for regularization following [7];

(b) we consider the mixup augmentation method, and we choose $\alpha = 0.2$ in the Beta distribution for mixup training;

(c) we use 5 epochs to gradually warm up learning rate at the beginning of the training; and

(d) we use the cosine learning rate schedule within 200 epochs, and set the initial learning rate as 0.1.

It is worth pointing out that our training setting above is selected based on training efficiency and does not equal to any setting reported in [7].

We refine the architecture by making the following adjustments to the original ResNets, and the resulting network is often called ResNet-D [7].

(a) The stride sizes are switched for the first two convolutions in the residual path of the down-sampling blocks.

(b) The 7x7 convolution in the stem is replaced by three smaller 3x3 convolutions.

(c) The stride-2 1x1 convolution in the skip connection path of the downsampling blocks is replaced by stride-2 2x2 average pooling and then a non-strided 1x1 convolution.

Table B.4: Network architectures of ResNets (left) and RLA-ResNets (right). Note that each "conv" layer shown in the table corresponds to the sequence BN-ReLU-Conv, except that the first "conv" layer corresponds to Conv-BN-ReLU. Varying the block counts $(B_1, B_2)$ gives rise to ResNet-50 and RLA-ResNet-50 $(B_1 = 4, B_2 = 6)$, ResNet-101 and RLA-ResNet-101 $(B_1 = 4, B_2 = 23)$, and ResNet-152 and RLA-ResNet-152 $(B_1 = 8, B_2 = 36)$.

| Output size | ResNet | RLA-ResNet |
|---|---|---|
| $112 \times 112$ | conv, $7 \times 7$, 64, stride 2 | |
| $56 \times 56$ | max pool, $3 \times 3$, stride 2 | |
| | $\begin{bmatrix} \text{conv}, 1 \times 1, 64 \\ \text{conv}, 3 \times 3, 64 \\ \text{conv}, 1 \times 1, 256 \end{bmatrix} \times 3$ | $\begin{bmatrix} \text{conv}, 1 \times 1, 64 \\ \text{conv}, 3 \times 3, 64 \\ \text{conv}, 1 \times 1, 256 \\ k = 32 \end{bmatrix} \times 3$ |
| $28 \times 28$ | $\begin{bmatrix} \text{conv}, 1 \times 1, 128 \\ \text{conv}, 3 \times 3, 128 \\ \text{conv}, 1 \times 1, 512 \end{bmatrix} \times B_1$ | $\begin{bmatrix} \text{conv}, 1 \times 1, 128 \\ \text{conv}, 3 \times 3, 128 \\ \text{conv}, 1 \times 1, 512 \\ k = 32 \end{bmatrix} \times B_1$ |
| $14 \times 14$ | $\begin{bmatrix} \text{conv}, 1 \times 1, 256 \\ \text{conv}, 3 \times 3, 256 \\ \text{conv}, 1 \times 1, 1024 \end{bmatrix} \times B_2$ | $\begin{bmatrix} \text{conv}, 1 \times 1, 256 \\ \text{conv}, 3 \times 3, 256 \\ \text{conv}, 1 \times 1, 1024 \\ k = 32 \end{bmatrix} \times B_2$ |
| $7 \times 7$ | $\begin{bmatrix} \text{conv}, 1 \times 1, 512 \\ \text{conv}, 3 \times 3, 512 \\ \text{conv}, 1 \times 1, 2048 \end{bmatrix} \times 3$ | $\begin{bmatrix} \text{conv}, 1 \times 1, 512 \\ \text{conv}, 3 \times 3, 512 \\ \text{conv}, 1 \times 1, 2048 \\ k = 32 \end{bmatrix} \times 3$ |
| $1 \times 1$ | global average pool, 1000-d $fc$, softmax | |

## B.3 Object detection and instance segmentation on MS COCO

**Implementation details** To show the transferability and the generalization ability, we experiment our RLA-Nets on the object detection task using three typical object detection frameworks: Faster R-CNN [11], Mask R-CNN [6] and RetinaNet [10]. For Mask R-CNN, we also show instance segmentation results. ResNet-50 and ResNet-101, pretrained on ImageNet, along with FPN [9] are used as backbone models. All detectors are implemented by open source MMDetection toolkit [1]. [2] We employ the default settings to finetune our RLA-Nets on COCO train2017 set, and evaluate the performance on COCO val2017 set. Specifically, the shorter side of input images are resized to 800. We train all detectors within 12 epochs using SGD with weight decay of 1e-4, momentum of 0.9 and mini-batch size of 8. The learning rate is initialized to 0.01 and is decreased by a factor of 10 after 8 and 11 epochs, respectively, i.e., the 1x training schedule.

The results shown in Tables 5 and 6 are all obtained from the models pretrained with standard training pipelines on ImageNet. The models with our proposed RLA modules show strong advantages in transfer learning, which significantly improve both object detection and instance segmentation. Thus, we make a reasonable conjecture that the performances of these tasks could be further improved with the help of the pretrained models with training and architecture refinements on ImageNet, as the transfer learning results in [7].

## B.4 Ablation study

In this subsection, we report ablation experiments on CIFAR with ResNet-164 as the baseline network. As in the ablation study on ImageNet in the main paper, we compare RLA modules with the same set of variations. Besides, we also compare with its DenseNet variant, i.e., dual path network (DPN) [2]. And different values of RLA channel $k$ are examined as well.

Moreover, we present additional results and discussions on the parallelizability, throughput and weight-sharing of RLA-Net in Sections B.4.4 and B.4.5.

---

[2]License: Apache License 2.0

Table B.5: Classification errors on the CIFAR-10 test set using ResNet-164 as the baseline model.

| Model | Params | Error (%) |
|---|---|---|
| ResNet-164 | 1.72M | $5.72 \pm 0.02$ |
| - channel +12 | 1.93M | 5.27 |
| - RLA-v1 ($k = 12$, unshared) | 1.90M | 5.17 |
| - RLA-v1 ($k = 12$, no exchange) | 1.72M | 5.51 |
| - RLA-v1 ($k = 12$, ConvLSTM) | 1.77M | 5.36 |
| - RLA-v1 ($k = 12$, PostAct.) | 1.74M | 5.39 |
| - DenseNet ($k = 12$, DPN) | 2.13M | 5.99 |
| - **RLA-v1** ($k = 12$) | 1.74M | **4.95** |
| - RLA-v2 ($k = 12$) | 1.74M | 5.53 |
| - RLA-v3 ($k = 12$) | 1.74M | 5.13 |
| - RLA-v4 ($k = 12$) | 1.74M | 5.35 |
| - RLA-v5 ($k = 12$) | 1.74M | 5.00 |
| - RLA-v6 ($k = 12$) | 1.74M | 5.62 |

Table B.6: Network architectures of ResNet-164 (left), ResNet-164 (channel+12) (middle) and RLA-ResNet-164 (right). Note that each "conv" layer shown in the table corresponds to the sequence BN-ReLU-Conv, except that the first "conv" layer corresponds to Conv-BN-ReLU.

| Output size | ResNet-164 | ResNet-164 (channel+12) | RLA-ResNet-164 |
|---|---|---|---|
| $32 \times 32$ | conv, $3 \times 3$, 16 | | |
| $32 \times 32$ | $\begin{bmatrix} \text{conv}, 1 \times 1, 16 \\ \text{conv}, 3 \times 3, 16 \\ \text{conv}, 1 \times 1, 64 \end{bmatrix} \times 18$ | $\begin{bmatrix} \text{conv}, 1 \times 1, 16 + 3 \\ \text{conv}, 3 \times 3, 16 + 3 \\ \text{conv}, 1 \times 1, 64 + 12 \end{bmatrix} \times 18$ | $\begin{bmatrix} \text{conv}, 1 \times 1, 16 \\ \text{conv}, 3 \times 3, 16 \\ \text{conv}, 1 \times 1, 64 \\ k = 12 \end{bmatrix} \times 18$ |
| $16 \times 16$ | $\begin{bmatrix} \text{conv}, 1 \times 1, 32 \\ \text{conv}, 3 \times 3, 32 \\ \text{conv}, 1 \times 1, 128 \end{bmatrix} \times 18$ | $\begin{bmatrix} \text{conv}, 1 \times 1, 32 + 3 \\ \text{conv}, 3 \times 3, 32 + 3 \\ \text{conv}, 1 \times 1, 128 + 12 \end{bmatrix} \times 18$ | $\begin{bmatrix} \text{conv}, 1 \times 1, 32 \\ \text{conv}, 3 \times 3, 32 \\ \text{conv}, 1 \times 1, 128 \\ k = 12 \end{bmatrix} \times 18$ |
| $8 \times 8$ | $\begin{bmatrix} \text{conv}, 1 \times 1, 64 \\ \text{conv}, 3 \times 3, 64 \\ \text{conv}, 1 \times 1, 256 \end{bmatrix} \times 18$ | $\begin{bmatrix} \text{conv}, 1 \times 1, 64 + 3 \\ \text{conv}, 3 \times 3, 64 + 3 \\ \text{conv}, 1 \times 1, 256 + 12 \end{bmatrix} \times 18$ | $\begin{bmatrix} \text{conv}, 1 \times 1, 64 \\ \text{conv}, 3 \times 3, 64 \\ \text{conv}, 1 \times 1, 256 \\ k = 12 \end{bmatrix} \times 18$ |
| $1 \times 1$ | global average pool, 10-d or 100-d $fc$, softmax | | |

### B.4.1 Comparison of RLA module and its variants

The results are reported in Table B.5. Again, we find that sharing the convolutions are indeed effective, and the improvement in accuracy is not fully caused by the increment in the network width. As DPN is not originally implemented for CIFAR and ResNet-164, we modified ResNet-164 following [2]. To our surprise, our implementation of DPN does not perform well in this experiment. For other variants, we also obtain consistent observations with the conclusions in the main paper. Detailed network architectures are introduced below.

**ResNet-164 (channel+12)**   For easy comparison, we repeat Table B.2 here with an additional column describing the structure of ResNet-164 (channel+12); see Table B.6.

**ConvLSTM based RLA**   A diagram for ConvLSTM based RLA module is shown in Figure B.2. Different from [13], our ConvLSTM cell is the convolutional counterpart of the conventional LSTM cell, instead of the fully connected LSTM (FC-LSTM) cell. All the cell updates are exactly the same as those in the Keras official implementation of the class `ConvLSTM2DCell`.

**Post-Activated RLA**   Motivated by [5], we experimented with the post-activated RLA module, see Figure B.3. Compared with Figure 3 in the main text, we only change the sequence BN-tanh-Conv to Conv-BN-tanh. It turns out that pre-activated RLA performs better on the pre-activated ResNet-164 model on CIFAR-10. In case of post-activated CNN-based models, post-activated RLA may still be worth trying.

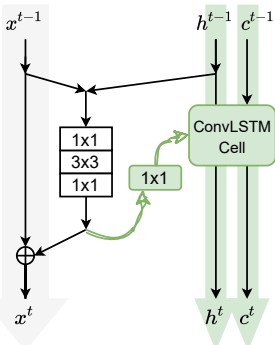

Figure B.2: Diagram of the ConvLSTM variant of the recurrent layer aggregation (RLA) module.

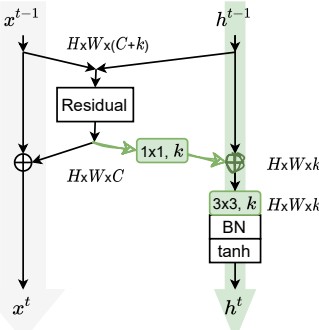

Figure B.3: Diagram of the post-activated variant of the recurrent layer aggregation (RLA) module.

### B.4.2 Comparison of integration strategies

We compare different strategies to combine our RLA module with the main CNN. In addition to the proposed design, we consider five variants (v2–v6), as depicted in Figure 4 in the main text. The performance of the variants are reported in Table B.5. The results on CIFAR are consistent with those on ImageNet, further validating our perspectives and statements in the main text.

### B.4.3 RLA channel $k$

RLA channel $k$ is defined as the number of filters in the shared 3x3 convolution in the RLA module. This hyperparameter allows us to vary the capacity and computational cost of the RLA module in a network. Based on RLA-v1, we experiment with various $k$ values in the set $\{8, 12, 16, 24\}$. Table B.7 shows that the performance is robust to different $k$ values, and the increase in complexity does not improve accuracy monotonically. Setting $k = 12$ strikes a good balance between accuracy and complexity. This result is consistent with the growth rate setting of DenseNet.

Table B.7: Classification error on CIFAR-10 using ResNet-164 with different RLA channel $k$.

| Model | Params | Error (%) |
|---|---|---|
| RLA-v1 ($k = 8$) | 1.73M | 4.98 |
| **RLA-v1** ($k = 12$) | 1.74M | **4.95** |
| RLA-v1 ($k = 16$) | 1.75M | 5.28 |
| RLA-v1 ($k = 24$) | 1.78M | 5.14 |

### B.4.4 Parallelizability and throughput of RLA-Net

Thanks to an anonymous reviewer's comment, we discuss the parallelizability and throughput of RLA-Net in this subsection. It is noticeable that adding RLA modules does not affect the paralleliz-

ability of the CNN because all existing parallelizable dimensions (batch, channel, width, height) are still parallelizable. Not like the commonly used RNNs in the literature, the proposed RLA module treats the layers of a deep CNN as inputs, while the feedforward procedure of a CNN is not parallelizable with respect to its depth. Thus, adding RLA does not affect the parallelizability of the resulting network, and the increment of complexity is similar to adding CNN modules, like SE- or ECA-block.

The throughput is measured as the number of images processed per second on a GPU. We present in Table B.8 some figures on the time cost of the proposed RLA module in the training and evaluation processes. Specifically, the evaluation is conducted on ImageNet validation set on 1x A100 GPU. Besides, we provide the training speed which has been conducted on 4x V100 GPUs previously. From the table below, it can be seen that adding SE, ECA or RLA module to ResNet50 costs about 22%, 21% or 26% more training time. Compared with the original ResNet50, introducing SE, ECA or RLA module costs about 0.2%, 0.9% and 2.8% more time when evaluating on ImageNet validation set.

Table B.8: Training and evaluation time of different modules using ResNet-50 as the backbone model.

| Model | Train (s/epoch) | Evaluation (ms/image) |
|---|---|---|
| ResNet-50 | 961 | 1.16 |
| +SE | 1171 | 1.17 |
| +ECA | 1161 | 1.17 |
| +RLA (Ours) | 1211 | 1.20 |

### B.4.5 Discussions on weight-sharing

Thanks to an anonymous reviewer's comment, we discuss why we propose weight-sharing in RLA in this subsection. Our proposed concept of layer aggregation contains a broad class of implementations including not only shared RLA but also its unshared version. Compared with shared RLA, the unshared version is more general but has more parameters.

We first explain why shared weights, i.e., the shared 1x1 and 3x3 Conv in Figure 3, are used at each layer of the RLA module. Weight-sharing is an important feature of the proposed RLA module, and the RLA path actually will become a deep CNN if unshared weights are used. While parameter sharing limits the expressiveness of a network, in practice, we have many examples where adding constraints helps. For example, convolution, as a regularized operation, incorporates our prior belief/inductive bias that contents in an image should be transition invariant. Similarly, weight-sharing in our RLA module represents our inductive bias that more distant layers could be less dependent; see Figure 2 (left) for support. Besides, it forces the module to learn a pattern different from the main CNN, i.e., ResNet corresponds to the shared-ordinal pattern (see Eq. 10), and RLA corresponds to a different shared-lag pattern (see Eq. 9). If the module is unshared, both the main CNN and the module learn the same shared-ordinal pattern, which possibly leads to redundancy and deteriorates the performance. Thus, we hypothesize that parameter sharing forces the RLA module to take up a functionality different from the main CNN, which turns out to be more helpful for the overall performance.

This inductive bias is further supported by our ablation experiments in Section 4.4. We repeat the results of these experiments in Table B.9. To provide more experiment results to support our heuristic, we also conduct experiments using ResNet-101 on ImageNet. Specifically, we have spotted advantages of our proposed RLA over its unshared variant in terms of both accuracy (+0.22%) and parameter count (-10%) on the ResNet-164 backbone on the CIFAR-10 dataset. Similar phenomena can be observed on ImageNet, where our proposed RLA module achieves +0.10% and +0.28% higher accuracy with 2% and 3% fewer parameters, respectively. Compared with ResNet-50, the differences on ResNet-101 are more significant. This can be explained that the 3rd stage of ResNet-101 is much longer, leading to the larger differences between shared and unshared versions. Specifically, for ResNet-101, the 3rd stage has 23 residual blocks, while it only has 6 residual blocks for ResNet-50. These consistent results in Table B.9 show that the inductive bias embedded in parameter-sharing is beneficial in terms of the trained model.

Table B.9: Comparisons of shared RLA and its unshared version.

| Data | Model | Params | FLOPs | Top-1 acc. | Top-5 acc. |
|------|-------|--------|-------|------------|------------|
| CIFAR-10 | RLA-ResNet-164 (Ours) | 1.74M | 8.74M | 95.05 | - |
|  | -Unshared variant | 1.90M | 8.74M | 94.83 | - |
| ImageNet | RLA-ResNet-50 (Ours) | 24.67M | 4.17G | 77.17 | 93.42 |
|  | -Unshared variant | 25.12M | 4.17G | 77.07 | 93.39 |
|  | RLA-ResNet-101 (Ours) | 42.92M | 7.79G | 78.52 | 94.20 |
|  | -Unshared variant | 44.05M | 7.79G | 78.24 | 93.97 |

To recommend a specific architecture, we aim at providing an "optimized" structure based on our experiments. Our results show that there is no downside to using the recurrent version. That's why we mainly propose RLA.