# OpenReview forum: "Recurrence along Depth: Deep Convolutional Neural Networks with Recurrent Layer Aggregation"
_NeurIPS.cc/2021/Conference — NeurIPS 2021 Poster_

### Official Review · Reviewer_uWZq · 2021-07-13

**Rating:** 7
**Confidence:** 3

**Summary:**

The paper proposed a recurrent layer aggregation (RLA) structure. The RLA structure proposes a recurrent module to aggregating information from the previous layers. The proposal is inspired by an additive form of describing the information aggregations in deep neural network and its similarity to autoregressive models. The structure is compatible with many popular CNN models including the ResNets, Xception and MobileNet. Extensive sets of experiments have been conducted to demonstrate the effectiveness of the proposed method on different vision tasks.

**Limitations And Societal Impact:**

There’s no discussion of limitations or negative social impacts in the paper. In general, concerns about negative social impacts of vision models are related to security issues like vulnerability to adversarial attacks and privacy issues due to the abuse of these models.

**Main Review:**

Pros
1. The proposed RLA is a class of modules compatible with different popular neural networks and other methods like channel attention. Experiment shows consistent performance improvements on different backbone models and tasks over model without RLA.
2. The motivation of the proposed RLA based on time series model is strong and well supported by experiment results in Table 2 and Fig. 2.
3. The structure of the paper is well-organized and easy to follow.

Cons

There’re no significant shortcomings in this work but two additional comments that might help improve the paper.
1. The paper shows the robustness of performance improvements to different RLA vairants on cifar10 datasets. Extension of such experiments to ImageNet and MSCOCO with similar results would support this conclusion significantly better.
2. The purpose of the paragraphs about interpretation of densenet as channel attention and the connection between this interpretation and motivation of RLA are not clear.

**Time Spent Reviewing:**

3 hours

---

> ### Author Response · Authors · 2021-08-10
> **Our point-to-point responses to your review**
>
> Thanks very much for your encouraging words and valuable suggestions! We sincerely appreciate your time in evaluating our paper, and our point-to-point responses to your comments are given below.
>
> **Experiments on ImageNet and MSCOCO**
>
> Thanks very much for this suggestion! We have started running ablation studies on the ImageNet dataset with ResNet-50 as the backbone model. Currently available results are given below. In general, the results are consistent with the ones on CIFAR-10, for example, our proposed RLA-v1 performs the best among many variants including v2-v5. We will update some remaining ablation study results of RLA-v6 and ConvLSTM variants ideally before 20/08.
>
> And MSCOCO experiments can only be arranged after finishing all ImageNet ones, due to the need for ImageNet pretrained model weights. We will proceed to MSCOCO if time allows.
>
> Some experiment results of ablation study:
>
> | Model | Top-1 acc. | Top-5 acc. |
> | ----- | :----------: | :----------: |
> | ResNet-50 | 75.30 | 92.20 |
> | - channel +32 | 76.46 | 93.28 |
> | - RLA-v1 (k=32) | 77.17 | 93.42 |
> | - RLA-v1 (k=32, unshared) | 77.07 | 93.39 |
> | - RLA-v2 (k=32) | 76.76 | 93.39 |
> | - RLA-v3 (k=32) | 77.05 | 93.46 |
> | - RLA-v4 (k=32) | 76.64 | 93.28 |
> | - RLA-v5 (k=32) | 76.70 | 93.40 |
>
>
>
> **Connection with channel attention**
>
> The concept of layer aggregation and the proposed RLA are related to, but not directly motivated by, channel attention. We discuss their connections for the two purposes below:
>
> (1)	To relate the newly proposed concept with an existing one and improve our understanding of the layer aggregation, i.e., layer aggregation generalizes channel attention across layers; and
>
> (2)	To show that layer aggregation is not a substitute to channel attention modules, but complementary to them, which inspires us to add the proposed RLA modules to ECAnets later.
>
> Sorry that we forgot to highlight our purposes in the context and will revise accordingly.
>
> Besides, thanks very much for shedding light on how to write the limitations and societal impact. In our revisions, we will clarify that our work has not been tested on the security aspect, nor does it contain any measures on safeguarding privacy. We will call for attention to these aspects when applying our method to practice.
>
> We hope that these responses meet your expectation! If we are allowed to update our manuscript and supplementary materials during the discussion period, we sincerely welcome you to check our revisions. Please do let us know if you have any follow-up questions or comments!

---

> > ### Comment · Reviewer_uWZq · 2021-08-21
> > **Review update**
> >
> > Thanks for the additional experiment results and clarification of my concerns in your response. I will keep my rating as 7.

---

> > > ### Author Response · Authors · 2021-08-24
> > > **Thanks for your support!**
> > >
> > > Thanks for your support! We have done some new experiments according to one reviewer’s comments, which study the differences between the RLA and its unshared version. They are attached in case you are interested.
> > >
> > > | Data | Model | Top-1 acc. | Top-5 acc. | Params (M) | FLOPs |
> > > | ---- | ----- | :----------: | :----------: | :----------: | :---------: |
> > > | CIFAR-10 | RLA-ResNet-164 (Ours) | 95.05 | - | 1.74 | 8.74M |
> > > |  | -Unshared variant | 94.83 | - | 1.90 | 8.74M |
> > > | ImageNet | RLA-ResNet-50 (Ours) | 77.17 | 93.42 | 24.67 | 4.17G |
> > > |  | -Unshared variant | 77.07 | 93.39 | 25.12 | 4.17G |
> > > |  | RLA-ResNet-101 (Ours) | 78.52 | 94.20 | 42.92 | 7.79G |
> > > |  | -Unshared variant | 78.24 | 93.97 | 44.05 | 7.79G |

---

> ### Author Response · Authors · 2021-08-20
> **Update of ablation study on ImageNet**
>
> We have finished the remaining ablation study on ImageNet. Please kindly check the updated results as below. Our proposed model is in bold. The new results are also consistent with the ones on CIFAR-10.
>
> | Model | Top-1 acc. | Top-5 acc. |
> | ----- | :----------: | :----------: |
> | ResNet-50 | 75.30 | 92.20 |
> | - channel +32 | 76.46 | 93.28 |
> | - **RLA-v1 (k=32)** | **77.17** | **93.42** |
> | - RLA-v1 (k=32, unshared) | 77.07 | 93.39 |
> | - RLA-v1 (k=32, [39] variant) | 76.58 | 93.10 |
> | - RLA-v1 (k=32, PostAct.) | 76.89 | 93.35 |
> | - RLA-v1 (k=32, ConvLSTM) | 77.08 | 93.47 |
> | - RLA-v2 (k=32) | 76.76 | 93.39 |
> | - RLA-v3 (k=32) | 77.05 | 93.46 |
> | - RLA-v4 (k=32) | 76.64 | 93.28 |
> | - RLA-v5 (k=32) | 76.70 | 93.40 |
> | - RLA-v6 (k=32) | 76.50 | 93.18 |

---

### Official Review · Reviewer_jn5x · 2021-07-15

**Rating:** 6
**Confidence:** 4

**Summary:**

This paper proposes a module called Recurrent Layer Aggregation (RLA) to improve the performance of general CNN backbones. This module is motivated by the analysis of both layer aggregation and the time series method. Experiments on tasks like classification, detection, and segmentation validate the effectiveness of this RLA module.

**Main Review:**

Originality:
The proposed RLA seems new to me, especially considering its relation with the time series analysis. The recurrent module is not simple usage of the off-the-shelf modules like LSTM or GRU but a custom one, which is interesting.

Quality:
The paper is technically sound. The proposed module is motivated by both layer aggregation and the time series method. The implementation of RLA is just a slight variant of the inspired formula Eq. (8), which is clean.
However, despite the custom recurrent module implementation, I think there should be an experimental comparison to show how the LSTM and GRU perform when they are used as the recurrent aggregation module (as Eq. 11). This will show us whether the custom recurrent module in the RLA is required or just an option.

Clarity:
There are some issues to address.
1. Line 138: I think there should be some more introductions about the autoregressive (AR) models and autoregressive moving average (ARMA) models, e.g. what they are handling, what their benefits are.
2. Line 116-129: Is there any specific reason to show the relationship between the layer aggregation and the channel attention? Does this motivate the proposed method?
3. Line 139: What is the meaning of the additive error $i^t$?
4. Line 191: Are the batch-norm layers share weights in the recurrent layers?

Significance:
I believe this work shows a certain level of significance as it highlights the importance of layer aggregation in a deep network, and the implementation of using a recurrent aggregation mechanism is new and simple.

**Time Spent Reviewing:**

4

---

> ### Author Response · Authors · 2021-08-10
> **Our point-to-point responses to your review**
>
> Thanks very much for your encouraging words and valuable suggestions! We sincerely appreciate your time in evaluating the paper, and our point-to-point responses to your comments are given below.
>
> **Proposed implementation vs. LSTM / GRU implementation**
>
> Thanks for this insightful comment! In fact, the proposed implementation is an “optimized” option based on our manual tuning within the scope introduced in Section 4.2 Ablation study.
>
> As you expected, LSTM and GRU can also be employed to implement recurrent layer aggregation modules. Exactly speaking, we can use *convolutional* versions of LSTM or GRU here since image data appear as high-dimensional tensors. We have already tried the ConvLSTM variant of the RLA module in our ablation study, see Table 3 for the result and Figure B.2 for the diagram of our ConvLSTM implementation. The observation is that ConvLSTM-based RLA
> - can improve the performance of ResNet-164 on CIFAR-10 (+0.36%).
> - requires more parameters and cannot outperform the proposed implementation (-0.41%).
>
> Besides, we are planning to test ConvLSTM and/or ConvGRU variants on ResNet-50 using ImageNet but are still waiting for the computing resources. We will try to update the results by 20/08.
>
> **Clarity**
>
> Thanks for letting us know these questions, and they can lead to an essential improvement of the quality of our paper!
> 1.	*More introductions to AR and ARMA models*
>
> Thanks for this suggestion! We are ready to add more introductions to AR and ARMA models.
>
> Simply speaking, in the literature of time series analysis, AR and ARMA models are most commonly used to fit and to predict time series data. For example, given the historical traffic volume $x^1, x^2, …, x^{t-1}$ on a specific road, the government wants to predict future traffic $ x^{t}, x^{t+1}, … $. Generally speaking, each observation $ x^t $ can depend on all previous ones in a nonlinear manner and the relationship may vary as $ t $ differs. AR and ARMA models are linear and assume a formulation where coefficients only depend on the *time lag* between the target and the past observation, shared across different target timestamp $ t $. Thus, they are simple, parsimonious (= small parameter count) and were proved to be useful in many applications.
>
> 2.	*Why we discuss channel attention*
>
> The concept of layer aggregation and the proposed RLA are related to, but not directly motivated by, channel attention. We discuss their connections for the two purposes below:
>
> (1)	To relate the newly proposed concept with an existing one and improve our understanding of the layer aggregation, i.e., layer aggregation generalizes channel attention across layers; and
>
> (2)	To show that layer aggregation is not a substitute to channel attention modules, but complementary to them, which inspires us to add the proposed RLA modules to ECAnets later.
>
> Sorry that we forgot to highlight our purposes in the context and are ready to do some revisions accordingly.
>
> 3.	*Additive error*
>
> Additive errors are included in time series models to represent randomness that we cannot and/or do not want to model. Without the additive error, the relationship $ x^t = 0.5 x^{t-1} $ defines a geometric sequence and becomes deterministic. With the additive error, for example, an AR(1) model $ x^t = 0.5 x^{t-1} + i^t $ means that our target $ x^t $ is positively correlated with the previous observation $ x^{t-1} $ (characterized by the positive coefficient 0.5) and its actual observation will subject to a random perturbation $ i^t $, which is unknown before time $ t $ and revealed after time $ t $. Thus, additive errors allow us to model sequences with randomness.
>
> We are ready to add more introduction on additive errors, as well as AR and ARMA models at Item 1.
>
> 4.	*Are the batch-norm layer weights shared*
>
> No, we do not share the weights of the batch normalization layers. In all diagrams in the paper, we intended to use plain rectangles to represent unshared operations and shaded round-corner rectangles for shared operations. Sorry that we forgot to illustrate this point in the main paper. We will add a legend to Figure 3 or describe it around line 191.
>
> We hope that these responses meet your expectation! If we are allowed to update our manuscript and supplementary materials during the discussion period, we sincerely welcome you to check our revisions. Please do let us know if you have any follow-up questions or comments!

---

> > ### Comment · Reviewer_jn5x · 2021-08-16
> > **After rebuttal review**
> >
> > Thanks for the authors' response. Most of my concerns are resolved, and I will keep my rating positive.

---

> > > ### Author Response · Authors · 2021-08-24
> > > **Thanks for your support!**
> > >
> > > Thanks for your support! We have done some new experiments according to one reviewer’s comments, which study the differences between the RLA and its unshared version. They are attached in case you are interested.
> > >
> > > | Data | Model | Top-1 acc. | Top-5 acc. | Params (M) | FLOPs |
> > > | ---- | ----- | :----------: | :----------: | :----------: | :---------: |
> > > | CIFAR-10 | RLA-ResNet-164 (Ours) | 95.05 | - | 1.74 | 8.74M |
> > > |  | -Unshared variant | 94.83 | - | 1.90 | 8.74M |
> > > | ImageNet | RLA-ResNet-50 (Ours) | 77.17 | 93.42 | 24.67 | 4.17G |
> > > |  | -Unshared variant | 77.07 | 93.39 | 25.12 | 4.17G |
> > > |  | RLA-ResNet-101 (Ours) | 78.52 | 94.20 | 42.92 | 7.79G |
> > > |  | -Unshared variant | 78.24 | 93.97 | 44.05 | 7.79G |

---

> ### Author Response · Authors · 2021-08-20
> **Update of ablation study on ImageNet**
>
> We have finished the ablation study on the ConvLSTM variant. Please kindly check the following table for the full results of our ablation study on ImageNet. Our proposed model is in bold. In general, the performances are consistent with those in Section 4.2. And the ConvLSTM variant achieves slightly worse Top-1 accuracy, but slightly better Top-5 accuracy, compared with our proposed structure.
>
> | Model | Top-1 acc. | Top-5 acc. |
> | ----- | :----------: | :----------: |
> | ResNet-50 | 75.30 | 92.20 |
> | - channel +32 | 76.46 | 93.28 |
> | - **RLA-v1 (k=32)** | **77.17** | **93.42** |
> | - RLA-v1 (k=32, unshared) | 77.07 | 93.39 |
> | - RLA-v1 (k=32, [39] variant) | 76.58 | 93.10 |
> | - RLA-v1 (k=32, PostAct.) | 76.89 | 93.35 |
> | - RLA-v1 (k=32, ConvLSTM) | 77.08 | 93.47 |
> | - RLA-v2 (k=32) | 76.76 | 93.39 |
> | - RLA-v3 (k=32) | 77.05 | 93.46 |
> | - RLA-v4 (k=32) | 76.64 | 93.28 |
> | - RLA-v5 (k=32) | 76.70 | 93.40 |
> | - RLA-v6 (k=32) | 76.50 | 93.18 |

---

### Official Review · Reviewer_Amr2 · 2021-07-16

**Rating:** 5
**Confidence:** 4

**Summary:**

A CNN architecture with two network paths (one is non-recurrent and another is recurrent) is introduced in this paper. The features from the two paths are aggregated through the RLA module (with residual connection) and redistributed to the individual paths. This brings performance improvements on image classification, object detection, instance segmentation at very small increases of parameter and computation costs.

**Limitations And Societal Impact:**

yes

**Main Review:**

Strengths
+ Lightweight design and works well with modern CNN architectures.
+ Good discussion of existing work.
+ Comprehensive experiments on several tasks with good results. RLA can be combined with ECA to achieve stronger performance.

Weaknesses
- The writing for the motivation behind the design of RLA is overly lengthy and not easy to follow. This part does not have major implications on RLA-Net as a whole yet it occupies a large portion of the paper. It is only relevant to Eq.8 in Sec 3.3 which is similar to vanilla RNN (already well known to the community).
- "applying RLA modules to CNNs is more than a simple combination of RNN and CNN (e.g., [39])" - Since RLA is different from [39] in terms of mutual information exchange and spatial feature preservation, there should be experiments showing the effects of these differences. Otherwise, it would be hard to justify the technical significance of RLA.
- The motivation behind using recurrent/shared weights for the second network path is not clear. There is nothing a recurrent network can model that a non-recurrent network cannot model. It can be simply replaced by a non-recurrent network. It seems to me that using recurrent/shared weights is just for keeping the parameter counts low.

Minor issue
Line 132: "DensNets" -> "DenseNets"

**Time Spent Reviewing:**

6

---

> ### Author Response · Authors · 2021-08-10
> **Our point-to-point responses to your review**
>
> Thanks very much for your valuable suggestions and comments! We sincerely appreciate your time in evaluating our paper, and our point-to-point responses to your comments are given below.
>
> **Writing for the motivation**
>
> Thanks for your careful reading! Yes, in Sections 3.1 & 3.2, only Eq. (8) is directly related to the proposed RLA module, but the other parts are also necessary. Please allow us to elaborate on it below.
>
> First, motivated by DenseNet, this paper attempts to look for a new design, which can extract features at the current layer more efficiently by reusing the information from previous layers. To this end, the concept of layer aggregation is introduced in Section 3.1, and it acts as a general framework for our discussion.
>
> Second, under the framework of layer aggregation, Eq. (8) is not the unique choice, and ResNet represents another direction for simplifying the DenseNet via parameter sharing. As a result, Section 3.2 first motivates Eq. (8) from viewpoints of time series analysis and RNNs, and a small experiment is also conducted to demonstrate that the mechanism of Eq. (8) will match real datasets better.
>
> Finally, the connection between channel attention and layer aggregation is discussed in Section 3.1 with two purposes:
>
> (1)	To relate the newly proposed concept with an existing one and improve our understanding of the layer aggregation, i.e., layer aggregation generalizes channel attention across layers; and
>
> (2)	To show that layer aggregation is not a substitute to channel attention modules, but complementary to them, which inspires us to add the proposed RLA modules to ECAnets later.
>
> The above is our explanation for Sections 3.1 & 3.2 from the authors’ viewpoints. From readers’ viewpoints, we may certainly miss, or unnecessarily emphasize, some things. So please do let us know your thoughts in more detail. For example, which parts are unnecessary in the main paper? Which points can help more if added to the main paper? We are eager for your inputs!
>
> **Comparison with [39]**
>
> Thanks for your careful reading! Following your suggestions, we have compared our design with the one in [39] on CIFAR-10, CIFAR-100 and ImageNet. For the sake of comparison, we have adopted a [39]-based variant by removing the information flow from the RLA path to the main CNN, while information flow from the main CNN to the RLA path is kept. The final state of the RLA path is only utilized in the fully connected layer, i.e., the classifier.
>
> Experiment results are provided in the table below. To conclude, the deviation of our design from the classical RNN structure is consistently beneficial, leading to 0.56% improvement on CIFAR-10, 0.76% on CIFAR-100 and 0.59% improvement in Top-1 accuracy on ImageNet.
>
> | Data | Model | Top-1 | Top-5 |
> | :---- | :----- | :----: | :----: |
> | CIFAR-10 | RLA-ResNet-164 (Ours) | 95.05 | - |
> |   | [39]-based variant | 94.49 | - |
> | CIFAR-100 | RLA-ResNet-164 (Ours) | 76.22 | - |
> |   | [39]-based variant | 75.46 | - |
> | ImageNet | RLA-ResNet-50 (Ours) | 77.17 | 93.42 |
> |   | [39]-based variant | 76.58 | 93.10 |
>
>
> **Motivation behind recurrence**
>
> We have carefully read your comments, and have reached two different understandings of your concerns. The first one is whether the shared weights at each layer of the RLA module can be replaced by unshared ones, and the second is whether the whole recurrent structure of RLA modules can be replaced by a non-recurrent version. In case that we may miss your points, we will answer both of them below.
>
> *(i) Weight-sharing*
>
> We first explain why the shared weights (i.e. the 1x1 and 3x3 conv at Figure 3) are used at each layer of the RLA module.
>
> Weight-sharing is an important feature of the proposed RLA module, and the RLA path actually will become a deep CNN if unshared weights are used. We agree with you that parameter sharing limits the expressiveness of a network. But in practice, we have many examples where adding constraints helps. For example, the convolution operation can be rewritten as matrix multiplication applied to vectorized images, where the coefficient matrix contains many zeros and shared kernel weights. There is nothing a CNN can model that a fully connected network (say, with residual connections) cannot. But CNNs still dominate computer vision tasks now. Convolution, as a regularized operation, incorporates our prior belief/inductive bias that, for example, contents in an image should be transition invariant.
>
> Similarly, weight-sharing in our RLA module represents our inductive bias that more distant layers could be less dependent; see Figure 2 (left) for support. This inductive bias is further supported by our ablation experiments in Section 4.2. Specifically, we have spotted advantages of our proposed RLA over its unshared variant in terms of both accuracy (+0.22%) and parameter count (-10%) on the ResNet-164 backbone on the CIFAR-10 dataset; see Table 3 for details. To provide more experiment results to support our heuristic, we newly conduct experiments using ResNet-50 on ImageNet. Similar phenomena can be observed, where our proposed RLA module achieves +0.10% higher accuracy with 2% fewer parameters. These consistent results show that the inductive bias embedded in parameter-sharing is beneficial in terms of the trained model. Please check the detailed experiment results below:
>
> | Data | Model | Top-1 | Top-5 |
> | :---- | :----- | :----: | :----: |
> | CIFAR-10 | RLA-ResNet-164 (Ours) | 95.05 | - |
> |   | Unshared variant | 94.83 | - |
> | ImageNet | RLA-ResNet-50 (Ours) | 77.17 | 93.42 |
> |   | Unshared variant | 77.07 | 93.39 |
>
>
>
> *(ii) The whole recurrent structure of RLA modules*
>
> We next explain below what will happen if the whole recurrent structure of RLA modules is replaced by a non-recurrent version.
>
> The proposed RLA module is actually motivated by DenseNet, which is a typical example of layer aggregation mechanism. Specifically, DenseNet uses a general (or non-recurrent) model to summarize the information from all previous layers, that is, obtaining previous information by skip connections directly, instead of rolling the information forward in a recurrent manner.
>
> Thus, without recurrent module structure, achieving layer aggregation (as in DenseNets) requires the number of parameters that is quadratic in the total number of layers. An unshared recurrent module structure reduces the order of parameter count to be linear in depth, and with parameter sharing, only a constant number of parameters is required regardless of depth.
> This difference in parameter counts also affects network design.
>
> Let us consider the RLA + ResNet combination. Roughly speaking, it will become an architecture similar to DenseNet if the recurrent structure in RLA is replaced by a non-recurrent version. We then compare the RLA + ResNet with DenseNet below.
>
> (1)	For DenseNet, a much larger proportion of parameters are used to summarize the information from all previous layers, and this actually is unavoidable since there are too many previous layers, especially for very deep CNNs. Correspondingly, less effort will be spent on extracting features at the current layer. In fact, DenseNet has been commonly criticized in the literature for its redundancy.
>
> (2)	Due to the recurrent structure, the proposed RLA is light-weighted. This guarantees that, for RLA + ResNet, the main part is ResNet, which concentrates on extracting features at the current layer, while the RLA is for further improvement.
>
> In short, the recurrent module design gives rise to a very light-weighted module for layer aggregation purpose, which can easily bring the benefit of layer aggregation to many CNN backbones, without the need to redesign or retune networks as in DenseNets or Dual Path Networks [7].
>
>
> We hope that these responses clarify your questions and concerns. If we are allowed to update our manuscript during the discussion period, we sincerely welcome you to check our revisions. Please do let us know if you have any follow-up questions or comments!

---

> > ### Comment · Reviewer_Amr2 · 2021-08-21
> > **Response**
> >
> > Thanks to the authors for the rebuttal. I find the work a little more convincing after reading the responses on the writing and experimental comparison. I think the improvements over [39] are not very significant although they are consistent. The recurrent structure seems to provide very little performance benefits over the non-recurrent/unshared version and I don't find that explaining it from the viewpoint of DenseNet is convincing (the unshared version does not need to do expensive layer aggregation as DenseNets do). The use of recurrent structure still seems not well-motivated in this paper.
> >
> > Based on the above, I would raise my rating by one point and encourage the authors to put good effort to improving the paper if accepted.

---

> > > ### Author Response · Authors · 2021-08-24
> > > **Further explanations on our RLA and its unshared version**
> > >
> > > Thanks very much for letting us know your thoughts!
> > > We would like to provide you with more experiment results and more explanations on why parameter sharing is adopted. To this end, we have newly accomplished the comparison based on ResNet-101 to further support our previous claims and observations.
> > >
> > >
> > > Our proposed concept of layer aggregation contains a broad class of implementations including not only shared RLA but also its unshared version. Compared with shared RLA, the unshared version is more general but has more parameters.
> > >
> > > (i)
> > >
> > > To recommend a specific architecture, we aim at providing an "optimized" structure based on our experiments. The following results show that there is no downside to using the recurrent version. That's why we mainly propose RLA.
> > >
> > > | Data | Model | Top-1 acc. | Top-5 acc. | Params (M) | FLOPs |
> > > | ---- | ----- | :----------: | :----------: | :----------: | :---------: |
> > > | CIFAR-10 | RLA-ResNet-164 (Ours) | 95.05 | - | 1.74 | 8.74M |
> > > |  | -Unshared variant | 94.83 | - | 1.90 | 8.74M |
> > > | ImageNet | RLA-ResNet-50 (Ours) | 77.17 | 93.42 | 24.67 | 4.17G |
> > > |  | -Unshared variant | 77.07 | 93.39 | 25.12 | 4.17G |
> > > |  | RLA-ResNet-101 (Ours) | 78.52 | 94.20 | 42.92 | 7.79G |
> > > |  | -Unshared variant | 78.24 | 93.97 | 44.05 | 7.79G |
> > >
> > >
> > >
> > > (ii)
> > >
> > > Our new results show that RLA-ResNet-101 outperforms its unshared variant by 0.28% and 0.23% in terms of top-1 and top-5 accuracies. Compared with ResNet-50, the differences on ResNet-101 are more significant. This can be explained that the 3rd stage of ResNet-101 is much longer, leading to the larger differences between shared and unshared versions. Specifically, for ResNet-101, the 3rd stage has 23 residual blocks, while it only has 6 residual blocks for ResNet-50.
> > >
> > > (iii)
> > >
> > > Our explanation for this surprising but consistent phenomenon is that weight-sharing represents our inductive bias that more distant layers could be less dependent. And it forces the module to learn a pattern different from the main CNN, i.e., ResNet corresponds to the shared-ordinal pattern (see Eq. 10), and RLA corresponds to a different shared-lag pattern (see Eq. 9). If the module is unshared, both the main CNN and the module learn the same shared-ordinal pattern, which possibly leads to redundancy and deteriorates the performance.
> > > Thus, we hypothesize that parameter sharing forces the RLA module to take up a functionality different from the main CNN, which turns out to be more helpful for the overall performance.
> > >
> > > Thanks for your comments again! We are ready to summarize the above differences between our RLA and its unshared version for better motivation.

---

> > > > ### Comment · Reviewer_Amr2 · 2021-09-02
> > > > **Shared version (recurrent design)**
> > > >
> > > > Thanks for reporting more results and providing the explanation.
> > > >
> > > > Although there are consistent performance improvement gains, they are too small to justify why the shared-weights/recurrent design should play a major role in RLA and the paper. Furthermore, the explanation provided in (iii) is merely an unverified hypothesis that cannot concretely justify the role of the shared-weights/recurrent design.
> > > >
> > > > Therefore, I still have my reservation and would prefer to keep the rating as it is.

---

> > > > > ### Author Response · Authors · 2021-09-04
> > > > > **shared-weight and unshared-weight designs**
> > > > >
> > > > > Please allow us to elaborate more about the shared-weight and unshared-weight designs.
> > > > >
> > > > > In order to understand how to make use of all previous layers’ information in extracting features at the current layer, we formally propose the concept of layer aggregation, while DenseNet is a typical example. On the other hand, under this framework, both shared-weight and unshared-weight designs are within our scope, and we finally decided to promote the share-weight module due to the following reasons:
> > > > >
> > > > > (1) Many currently used deep networks are already good enough in terms of performance, and this paper aims to provide an additional improvement from the aspect of layer aggregation.  Actually, we have no attempt in constructing a completely new architecture. Along the lines, in the running network, say ResNet + RLA, we expect that the main job is done by the ResNet, while the RLA is for improvement only. As a result, in our consideration, the proposed module should be very light-weighted and compatible with most mainstream deep networks.
> > > > >
> > > > > (2) The RLA path will become a deep CNN if unshared weights are used. As a result, the resulting architecture will have two paths: the main network path and the unshared-weight path, and they almost have the same mechanism. This does not match well our original consideration at Item (1).
> > > > >
> > > > > (3) The shared-weight design does have a better performance consistently although the improvement is small as you mentioned. More importantly, it is very light-weighted, and this exactly meets our need at Item (1).
> > > > >
> > > > > (4) It is impossible to verify whether the shared-weight or unshared-weight design has a better performance by using theorems or exhaustive experiments. But the idea of shared weights originates from RNN in machine learning and ARMA time series models in statistics, and our choice is mainly due to their overwhelming performance on sequential data in both areas.

---

> ### Author Response · Authors · 2021-08-20
> **Update of ablation study on ImageNet**
>
> We have finished the ablation study on ImageNet using ResNet50. Please kindly check the results below if you are interested. Our proposed model is in bold. In short, the performances are consistent with those reported in Section 4.2.
>
> | Model | Top-1 acc. | Top-5 acc. |
> | ----- | :----------: | :----------: |
> | ResNet-50 | 75.30 | 92.20 |
> | - channel +32 | 76.46 | 93.28 |
> | - **RLA-v1 (k=32)** | **77.17** | **93.42** |
> | - RLA-v1 (k=32, unshared) | 77.07 | 93.39 |
> | - RLA-v1 (k=32, [39] variant) | 76.58 | 93.10 |
> | - RLA-v1 (k=32, PostAct.) | 76.89 | 93.35 |
> | - RLA-v1 (k=32, ConvLSTM) | 77.08 | 93.47 |
> | - RLA-v2 (k=32) | 76.76 | 93.39 |
> | - RLA-v3 (k=32) | 77.05 | 93.46 |
> | - RLA-v4 (k=32) | 76.64 | 93.28 |
> | - RLA-v5 (k=32) | 76.70 | 93.40 |
> | - RLA-v6 (k=32) | 76.50 | 93.18 |

---

### Official Review · Reviewer_ngdy · 2021-07-16

**Rating:** 6
**Confidence:** 4

**Summary:**

This paper is inspired by the layer aggregation concept which is also used in DenseNet to better reuse the information from previous layers and extract features at the current layer. Then the authors propose a light-weighted module, called recurrent layer aggregation (RLA), and prove it is compatible with many common structures in ResNets, Xception and MobileNetV2. Finally, they use the controlled experiments to show the effectiveness of the RLA modules on classification and detection tasks.


**Ethical Concerns:**

This paper only did the ablation study on the CIFAR-10 dataset with ResNet-164 as the backbone network. Because the CIFAR-10 dataset is very small, while ResNet-164 is more capable for a larger dataset. There are maybe over-fitting situations and lead to bias in the ablation conclusion.

**Limitations And Societal Impact:**

The authors seem not to provide enough limitations and potential negative social impact of their work.
I suggest the authors to provide a description of the potential performance regression due to the introduction of the RLA module.


**Main Review:**

This paper has the good motivation to discuss better utilize the common information among adjacent layers to improve the feature extraction capability of the deep networks.
From the clarity aspect, I would suggest the authors to put the Figure B.4: (RLA modules with different types of connections) from the supplementary to the main paper. Just use the formula to express the module structure and its variants is not as clear as the images.
The main concern about this paper is the real performance during training and inference. Because of the introducing the recurrent layer aggregation module, layers need to also consider the previous states of adjacent layers. So similar to RNN modules, it is not friendly to the parallel acceleration like GPU device. I am curious about the real performance number when adding the RLA module.

**Time Spent Reviewing:**

5 hours

---

> ### Author Response · Authors · 2021-08-10
> **Our point-to-point responses to your review**
>
> Thanks very much for your encouraging words and valuable suggestions! We sincerely appreciate your time in evaluating the paper, and our point-to-point responses to your comments are given below.
>
> **Graphical illustration of module structure**
>
> This is a good suggestion, and we are ready to move Figure B.4 to the main paper!
>
> **Parallelizability**
>
> Simply speaking, adding RLA modules does not affect the parallelizability of the CNN because all existing parallelizable dimensions (batch, channel, width, height) are still parallelizable.
>
> Not like the commonly used RNNs in the literature, the proposed RLA module treats the layers of a deep CNN as inputs, while the feedforward procedure of a CNN is not parallelizable with respect to its depth. Thus, adding RLA does not affect the parallelizability of the resulting network, and the increment of complexity is similar to adding CNN modules, like SE- or ECA-block.
>
> **Real performance during training and inference (Performance regression)**
>
> Thanks for reminding us! We present below some figures on the time cost of the proposed RLA module at training and evaluation processes. The training process has been tested on 4x V100 GPUs previously, and the evaluation is newly conducted on ImageNet validation set on 1x A100 GPU. From the table below, it can be seen that adding SE, ECA or RLA module to ResNet50 costs about 22%, 21% or 26% more training time. Compared with the original ResNet50, introducing SE, ECA or RLA module costs about 0.2%, 0.9% and 2.8% more time when evaluating on ImageNet validation set.
>
> We are ready to incorporate the above description of the potential performance regression into the limitations and societal impact of our work. Please feel free to let us know whether the metrics in the table below suffice.
>
>
> | Model | Train (s/epoch) | Evaluation (s/50,000images) | Evaluation (ms/image) |
> | ----- | :------------------: | :---------------------------: | :---------------------: |
> | ResNet-50 | 961 | 58.23 | 1.16 |
> | +SE | 1171 | 58.36 | 1.17 |
> | +ECA | 1161 | 58.73 | 1.17 |
> | +RLA (Ours) | 1211 | 59.89 | 1.20 |
>
>
>
> **Ablation study**
>
> We agree with you that our current ablation study is only limited to one model and one small dataset (due to limited computational resources). Following your suggestions, in order to provide a more reliable comparison, we have started running more experiments on ImageNet using ResNet-50, aiming to better support our conclusions on the ablation study. Currently available results are given below. In general, the results are consistent with the ones on CIFAR-10, for example, our proposed RLA-v1 performs the best among many variants including v2-v5. We can update some remaining ablation study results of RLA-v6 and ConvLSTM variants ideally before 20/08.
>
> Some experiment results of ablation study:
>
> | Model | Top-1 acc. | Top-5 acc. |
> | ----- | :----------: | :----------: |
> | ResNet-50 | 75.30 | 92.20 |
> | - channel +32 | 76.46 | 93.28 |
> | - RLA-v1 (k=32) | 77.17 | 93.42 |
> | - RLA-v1 (k=32, unshared) | 77.07 | 93.39 |
> | - RLA-v2 (k=32) | 76.76 | 93.39 |
> | - RLA-v3 (k=32) | 77.05 | 93.46 |
> | - RLA-v4 (k=32) | 76.64 | 93.28 |
> | - RLA-v5 (k=32) | 76.70 | 93.40 |
>
>
> We hope that these responses clarify your questions and concerns. If we are allowed to update our manuscript during the discussion period, we sincerely welcome you to check our revisions. Please do let us know if you have any follow-up questions or comments!

---

> > ### Comment · Reviewer_ngdy · 2021-08-16
> > **Reply to authors' rebuttal**
> >
> > Thanks for your detailed responses to my comments. They solve some of my initial concerns. I keep my rating as the original.

---

> > > ### Author Response · Authors · 2021-08-24
> > > **Thanks for your support!**
> > >
> > > Thanks for your support! We have done some new experiments according to one reviewer’s comments, which study the differences between the RLA and its unshared version. They are attached in case you are interested.
> > >
> > > | Data | Model | Top-1 acc. | Top-5 acc. | Params (M) | FLOPs |
> > > | ---- | ----- | :----------: | :----------: | :----------: | :---------: |
> > > | CIFAR-10 | RLA-ResNet-164 (Ours) | 95.05 | - | 1.74 | 8.74M |
> > > |  | -Unshared variant | 94.83 | - | 1.90 | 8.74M |
> > > | ImageNet | RLA-ResNet-50 (Ours) | 77.17 | 93.42 | 24.67 | 4.17G |
> > > |  | -Unshared variant | 77.07 | 93.39 | 25.12 | 4.17G |
> > > |  | RLA-ResNet-101 (Ours) | 78.52 | 94.20 | 42.92 | 7.79G |
> > > |  | -Unshared variant | 78.24 | 93.97 | 44.05 | 7.79G |

---

> ### Author Response · Authors · 2021-08-20
> **Update of ablation study on ImageNet**
>
> We have finished the remaining ablation study on ImageNet. Please kindly check the updated results as below. Our proposed model is in bold. The new results are also consistent with the ones on CIFAR-10.
>
> | Model | Top-1 acc. | Top-5 acc. |
> | ----- | :----------: | :----------: |
> | ResNet-50 | 75.30 | 92.20 |
> | - channel +32 | 76.46 | 93.28 |
> | - **RLA-v1 (k=32)** | **77.17** | **93.42** |
> | - RLA-v1 (k=32, unshared) | 77.07 | 93.39 |
> | - RLA-v1 (k=32, [39] variant) | 76.58 | 93.10 |
> | - RLA-v1 (k=32, PostAct.) | 76.89 | 93.35 |
> | - RLA-v1 (k=32, ConvLSTM) | 77.08 | 93.47 |
> | - RLA-v2 (k=32) | 76.76 | 93.39 |
> | - RLA-v3 (k=32) | 77.05 | 93.46 |
> | - RLA-v4 (k=32) | 76.64 | 93.28 |
> | - RLA-v5 (k=32) | 76.70 | 93.40 |
> | - RLA-v6 (k=32) | 76.50 | 93.18 |

---

### Decision · Program_Chairs · 2021-09-27

**Decision:**

Accept (Poster)

**Comment:**

The paper presents a new design for aggregating information from shallower to deeper layers of a neural network (i.e., an alternative to ResNet or DenseNet-style connections).  Following the author response and discussion, 1 reviewer rating is negative, 3 are positive, with overall sentiment being marginally positive.  The paper presents a clear design and provides sufficient experimental evidence of benefits.  A downside is that performance benefits, even if consistent, are of an incremental nature over existing network designs.